# Multistage lithospheric drips control active basin formation within an uplifting orogenic plateau

A. Julia Andersen [1] ✉, Oguz Hakan Göğüş [2], Russell N. Pysklywec [1], Ebru Şengül Uluocak [3,4] & Tasca Santimano[1]

According to GNSS/INSAR measurements, the Konya Basin in Central Anatolia is undergoing rapid subsidence within an uplifting orogenic plateau. Further, geophysical studies reveal thickened crust under the basin and a fast seismic wave speed anomaly in the underlying mantle, in addition to a localised depression in calculated residual topography (down to 280 m) over the Konya Basin, based on gravity-topography considerations. Using scaled laboratory (analogue) experiments we show that the active formation of the Konya Basin may be accounted for by the descent of a mantle lithospheric drip causing local circular-shaped surface subsidence. We interpret that the Konya Basin is developing through a secondary drip pulse that is contemporaneous with broad plateau uplift caused by a larger-scale lithospheric drip since the Miocene. The research reveals that basin evolution and plateau uplift may be linked in a multistage process of lithospheric removal during episodic development of orogenic systems.

The geodynamic mechanisms of plateau evolution—such as in Tibet, Altiplano-Puna, and Anatolia—are tectonically complex, occurring in the hinterland of convergent orogenic systems. These plateaus have topographic, thermal, and geological characteristics that reflect an interplay of plate shortening and mantle dynamics[1–5]. Internally drained (deep) sedimentary basins of various scales have been described within the interior of these orogenic plateaus worldwide. However, the origin of such basins amidst an evolving orogeny and topographic rise of the plateau is not well resolved. Here, we explore the geodynamic origin of the Konya Basin, located in the interior of the Central Anatolian Plateau, as one such example of a closed basin-forming event within large-scale plateau uplift (Fig. 1a, b).

The Central Anatolian orogenic plateau is characterised by ~1.5–2 km average elevation with low relief and lower topography in the plateau interior, and high relief mountain ranges at its northern (Pontides) and southern (Taurides) margins (swath profile in Supp. Figure 1a). Cosmogenic dating of river terraces (Kızılırmak) in the plateau interior suggests surface uplift since ~2 Ma, although at much lower rates compared to the Pontides and Taurides[6,7]. Stratigraphic and geomorphological evidence determines an earlier onset for plateau uplift in the interior. For example, palaeontological evidence (e.g., mollusc bearing sections) from the Karaman Basin of the south-central plateau interior indicates that this region emerged from below sea level by at least ~11 Ma[7,8]. Furthermore, paleo-elevation estimates derived from incised valleys across basalt terraces in the Cappadocia Volcanic Province indicate surface uplift of up to ~1000 m since 8 Ma[9]. Meijers et al.[10] interprets[10] that gradual diminishing of (δ[18]O) oxygen stable isotope ratios from Miocene lacustrine carbonate deposits of continental basins within the plateau interior was caused by surface uplift (~11–5 Ma). Göğüş et al. suggests[3] that folding of the Kırşehir arc during the Eocene-Miocene led to lithospheric "dripping" and subsequently uplift of the Central Anatolian plateau.

While the eastern and western domains of Anatolia are affected by plate boundary activities, Central Anatolia is essentially tectonically inert. There are no known active crustal scale fault systems[11] or associated seismic activity[12] in the plateau interior except the NW-SE

[1]Department of Earth Sciences, University of Toronto, Toronto, ON, Canada. [2]Istanbul Technical University, Eurasian Institute of Earth Sciences, Istanbul, Turkey. [3]Department of Geophysical Engineering, Çanakkale Onsekiz Mart University, Çanakkale, Turkey. [4]Lithosphere Dynamics, GFZ German Research Centre for Geosciences, Potsdam, Germany. ✉e-mail: julia.andersen@mail.utoronto.ca

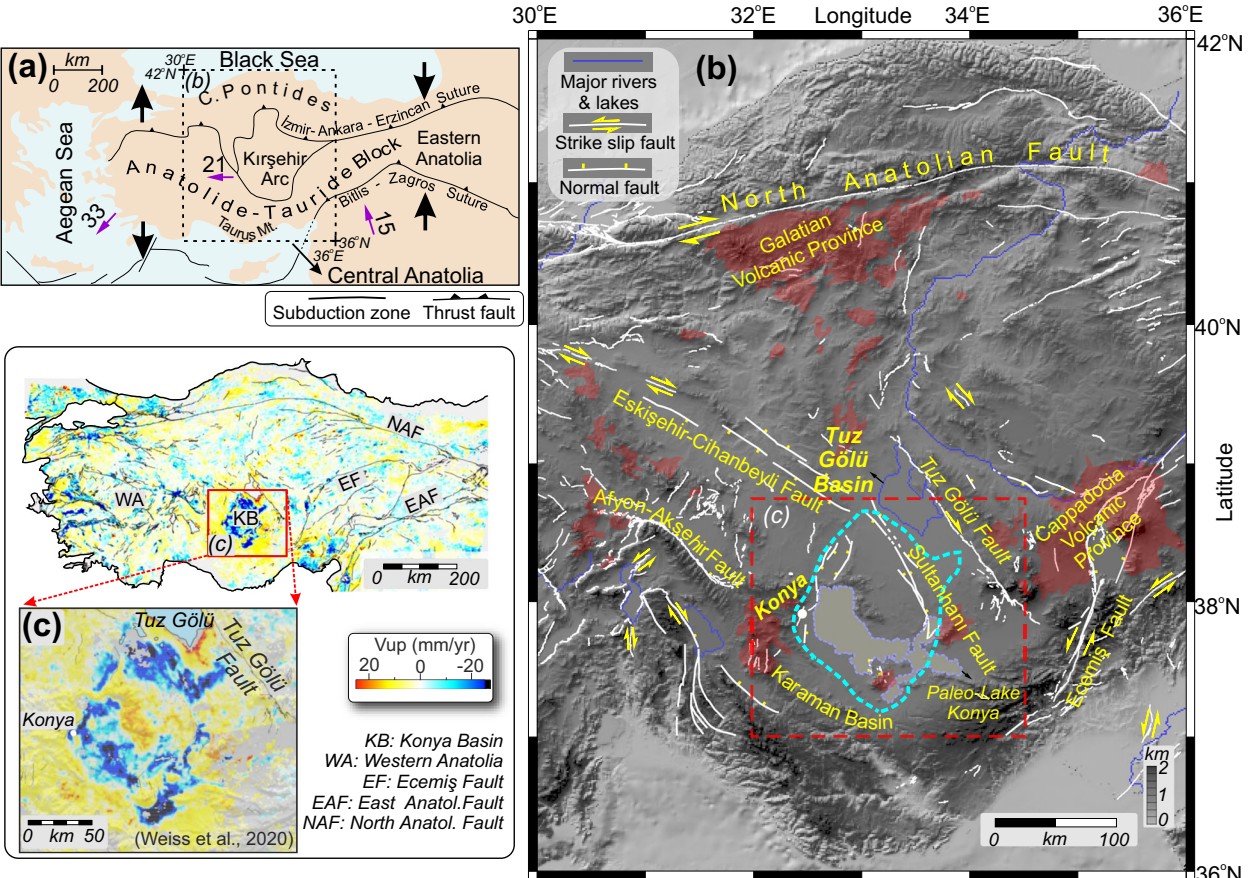

**Fig. 1 | Geological and tectonic setting of study area. a** Tectonic regions and plate motions of Anatolia. Purple arrows indicate relative plate motions with respect to Eurasia (numbers show the average velocities in mm/yr)[80–83]. **b** Shaded relief topographic map of Central Anatolia including volcanic provinces (Galatia and Cappadocia Volcanic Provinces shaded in red). The blue dashed line outlines the boundaries of the Konya Basin and a shaded area indicates paleolake Konya. White lines show main active faults in Anatolia with yellow arrows indicating the sense of shear[11,21]. The figure was created by the Generic Mapping Tools GMT 6[84]. **c** GNSS and INSAR based vertical velocities ($V_{up}$) in mm/yr in the Konya Basin showing rapid subsidence of the basin up to rates of >50 mm/yr[24].

trending Tuz Gölü fault (Fig. 1b). Based on geodetic measurements and geological interpretations, Özbey et al. interpreted[13] this fault system to operate primarily as a strike-slip shear zone, although it is encompassed within an overall region of low magnitude strain[14]. Fault plane solutions associated with the Bala earthquake sequences (2005–2008), ~50 km north of the Tuz Gölü fault, agree with the strike-slip characteristics of the seismically active shear zone with a complex multiphase evolution[15,16]. In addition, structural, stratigraphic, and sedimentological data from the Tuz Gölü fault have been used to interpret major motion of this fault system as initially strike-slip with a normal fault component between the post Paleogene to Pliocene[17], but this transitioned to a thrust fault during the Late Miocene to Early Pliocene[18] (Fig. 1b).

The Konya Basin is a circular shaped, intra-continental, closed (endorheic), sedimentary basin (Fig. 1c) that initially formed as a fore-arc basin in Central Anatolia following orogenic events between the Late Cretaceous and the Miocene epochs[19]. The basin preserves a record of long-term sedimentation owing to the lack of external drainage that would otherwise disperse sediments[10,20]. These sediments are lacustrine, and deposition continued until the Pleistocene based on a well-defined paleo shoreline of the paleolake Konya (Fig. 1b)[21–23].

Weiss et al. combined[24] Sentinel-1 InSAR and Global Navigation Satellite System (GNSS) measurements to map the present-day surface velocity and strain rate across the entire Anatolian plate. These data show a distinct bulls-eye pattern of active subsidence at the Konya Basin with vertical velocity rates approaching ≥20 mm/year (Fig. 1c).

However, as is consistent with the geological evidence described above, the measured internal deformation of the basin is minimal with low shear strain rates and lack of folding and/or active faulting[11,24]. These data from Weiss et al. present[24] a geodynamic puzzle: why is the Konya Basin rapidly subsiding amidst an overall uplifting, and tectonically quiet plateau?

Here, we combine scaled laboratory (analogue) experiments and analyses of geophysical and geological data to explore the dynamics of basin evolution within the Central Anatolian plateau interior. The topographic evolution of a 3D analogue model is measured to quantify the surface expression of lithospheric removal that may account for basin tectonics within the Alpine-Himalayan orogenic plateau system. Model results are interpreted within the framework of the geological evolution of the Central Anatolian plateau over the last ca. 10 My, and compared against the active subsidence of the Konya Basin, an enigmatic region of subdued seismic activity and muted crustal deformation. This research underscores what might be an overlooked multistage process of lithospheric removal within a large-scale orogenic system.

## Results

### Geophysical constraints on the evolution of the Konya Basin

In addition to the InSAR and GNSS measurements revealing active vertical displacement of the Konya Basin[24], there are several geophysical anomalies localised around the basin region (Fig. 2). Seismic studies by Vinnik et al. and Kind et al. suggest[25,26] a thin lithosphere

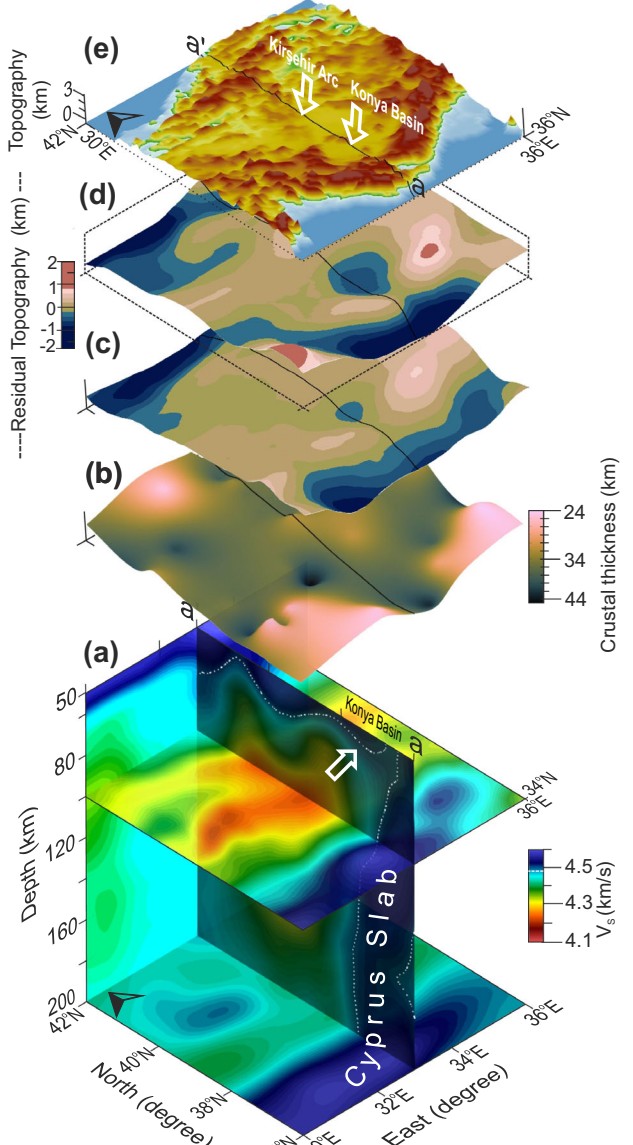

**Fig. 2 | Geophysical anomalies of Central Anatolia. a** S-wave seismic tomography along the N-S transect (a'-a) showing a fast seismic wave speed anomaly beneath the Konya Basin indicated by a white arrow, and another anomaly beneath the Kırşehir Arc to the north[27]. The white dashed line approximately delineates the lithosphere-asthenosphere-boundary (LAB). **b** Crustal thickness variations derived from Vanacore et al.[32]. **c** Our calculation of residual (non isostatic) topography showing negative residuals in the Konya Basin. **d** For comparison, we show residual topography estimates from Howell et al. [34]. **e** Surface topographic map of Central Anatolia[79].

beneath Central Anatolia with the lithosphere asthenosphere boundary (LAB) at 60–100 km depth. There is a fast seismic wave speed anomaly (the isotropic S-wave anomaly–$V_s$) beneath the Kırşehir arc (Fig. 2a, Fichtner et al.,[27]) that has been interpreted as a remnant of a large-scale drip event that uplifted the Central Anatolian plateau[3]. Furthermore, Fig. 2a shows that there is another fast seismic wave speed anomaly ($V_s \geq 4.4$ km/s[27]) between 50 km and 80 km depth, located directly beneath the Konya Basin indicating the presence of colder/denser lithosphere. This Konya anomaly is located north of the Cyprus slab and extends into the underlying mantle.

The Konya Basin lithosphere is characterised by a thick crust of ~40 km[28,29] with local thickening in the crust (up to 44 km) on the periphery of the basin (Fig. 2b) (ref. [28–32]). Crustal thickness variations

obtained from regional and large-scale data are shown along a 1° swath profile (shaded areas) centred at 33°E (solid lines) in Supp. Fig. 1b.

Mantle convective support of topography can be analyzed from elastically uncompensated long-wavelength gravity anomalies[33–36]. Based on the relationship between gravity and topography, it is generally suggested that the ratio associated with convective support at long-wavelengths (≥~300 km) is 50 mGal/km for continents and 30 mGal/km for ocean bathymetry (Howell et al.,[34] and references therein). By subtracting the isostatically supported topography based on crustal thickness[29] from observed topography, one can obtain the non-isostatic component of the topography (i.e., the residual topography, Fig. 2c[37]). Our calculations show a residual topography variation with a local depression (~280 m) across the Konya Basin (Fig. 2c, Supp. Fig. 1a). These findings are in concordance with residual topography anomaly estimates in the eastern Mediterranean from satellite gravity-topography (GOCE) data (Fig. 2d; Fig. 13c in Howell et al.,[34]). Figure 2e demonstrates that the surface topography of the Konya Basin is flat in comparison to the region surrounding the basin.

Based on geophysical anomalies in the region, we hypothesise that lithospheric dynamics account for active subsidence of the Konya Basin. Namely, the presence of a fast seismic wave speed anomaly (Fig. 2a) indicates a colder/denser lithospheric body sinking into the underlying sub-lithospheric mantle beneath the basin. Negative residual topography indicates that the topography in the region is non-isostatic, and dynamic topography is causing the surface to be substantially (~280 m) lower than it should be according to gravity-topography isostatic considerations. Together, these geophysical data may indicate the presence of a lithospheric drip actively pulling the crust downward. The geophysical anomalies are consistent with documented observables for lithospheric removal interpreted in other regions around the globe, such as the Tulare Lake basin of the southern Sierra Nevada[38], Lake Titicaca of the Altiplano plateau[39], and the Arizaro Basin of the Puna plateau[40].

We note that this hypothesis of active lithospheric dripping beneath the Konya Basin may present a complicated geodynamic framework in the context of the broad evolution of plateau uplift over the past ~10 My. If we consider that the Central Anatolian Plateau overall has been uplifting owing to a primary stage of lithospheric dripping[3,10,41], interpretation of the geophysical and geological data prompts the question whether it is possible to have a secondary, or late-stage dripping event causing active basin subsidence simultaneously? To reconcile these findings with the subsidence observed by Weiss et al.[24], we conducted 3D analogue modelling of lithosphere-mantle dynamics for the region. The geodynamic modelling investigates how a secondary event of lithospheric removal via dripping can cause basin subsidence within a regime of large-scale plateau evolution.

## Dynamics and the surface response to late stage dripping lithosphere

A physical scaled analogue model was setup (see Methods) to explore how gravitationally unstable lithosphere drips into the underlying mantle. An initial perturbation develops into a primary drip (Fig. 3a–c) that results in removal of mantle lithosphere as well as plateau uplift over the first ca. 40 h. Side-view images of the experiment show the primary drip pulse at 10 h (Fig. 3a) when the drip has started to descend, and by 25 h (Fig. 3b) the drip is sinking more rapidly and approaching the bottom of the tank. There is no visible deformation of the crust on the surface area directly above the drip (or elsewhere; Fig. 3c). We provide more detailed information of topographic evolution of the surface due to the primary drip in Supp. Fig. 3.

Notably, a secondary drip event (Fig. 3d–h) develops following the primary drip (similar to the phenomenon observed in Pysklywec & Cruden, (2004)). At 50.6 h after the primary drip reaches the bottom of

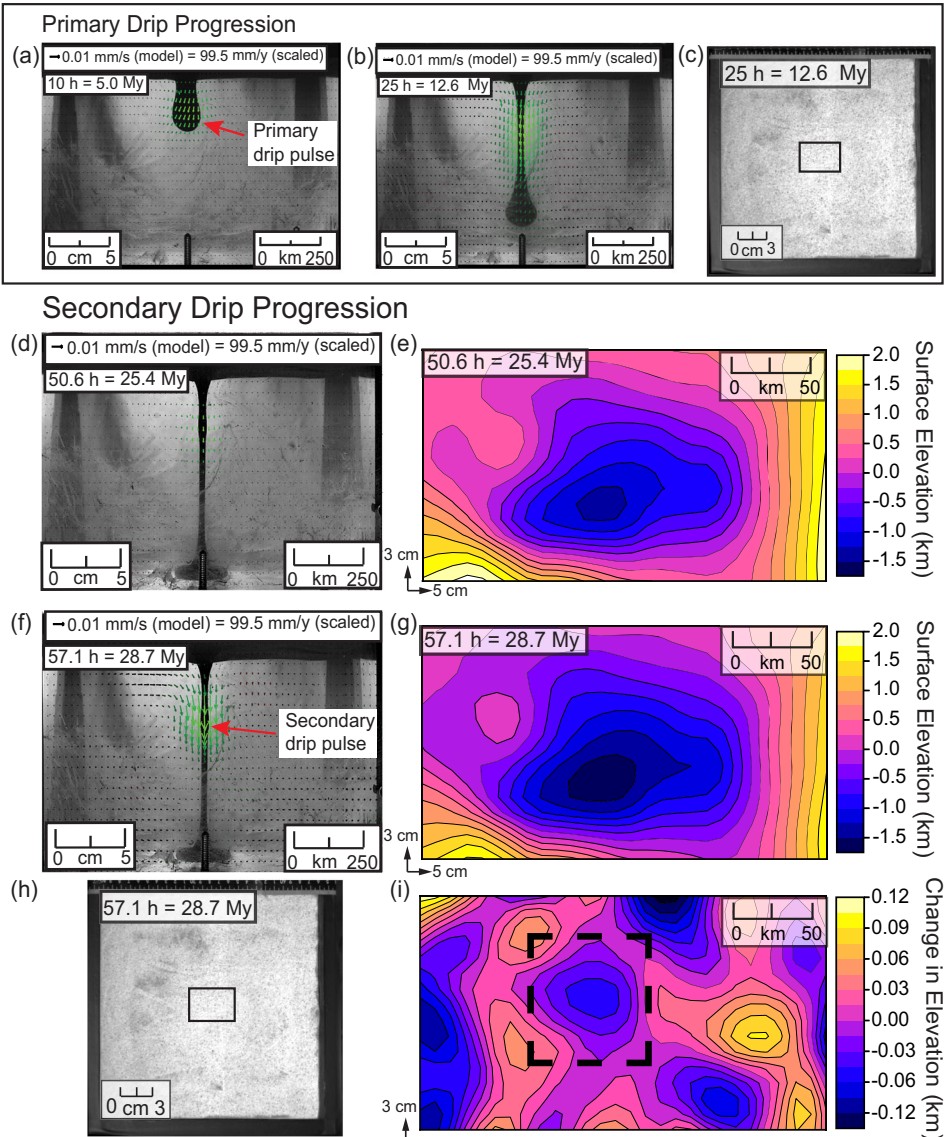

**Fig. 3 | Results of laboratory (analogue) experiments. a** Sideview image of the primary drip at 10 h. The drip has begun descending through the tank and is characterised by a bulbous drip head. **b** Sideview image of the primary drip at 25 h. The drip has almost touched the bottom of the box. The neck of the drip is thin (~1/4 the width of the drip head). Green vectors show increased downward velocity of the drip. **c** Digital image of the model top surface at 25 h. The rectangular box outlines a 3 × 5 cm surface area above the drip on the model surface. The image shows the unperturbed upper crust layer indicating that there is no horizontal crustal tectonic deformation—such as shortening or extension--reaching the surface despite the underlying drip behaviour. **d** Sideview image of the experiment at 50.6 h. There is a growth of a secondary drip, ~2 cm below the base of the lithosphere. **e** Surface

elevation contour map directly above the secondary drip at 50.6 h. A basin is visible in the surface that scales to 1.25 km deep. **f** Sideview image of drip progression 6.5 h later (57.1 h since the start) in the experiment. Now the secondary drip pulse has travelled ~3 cm beneath the lithosphere (green vectors). **g** Surface elevation contour map directly above the secondary drip at 57.1 h. The basin has deepened to 1.5 km. **h** A digital image of the model surface at 57.1 h. The sand crust layer does not show horizontal deformation at the surface while the lithosphere drips underneath (i.e., this is an asympomatic drip; Suppl. Fig. 2). **i** Vertical displacement of the model surface between 50.6 h and 57.1 h. The negative displacement in the dashed rectangular boundaries corresponds to a subsidence of the basin formed by the secondary drip.

the box and a period of low activity in the model, a secondary pulse of downwelling starts to develop within the stretching lithosphere ("drip tail") as indicated by the distribution of green velocity vectors (Fig. 3d). Note that the drip tail is thinned and elongated, but an exception is in the middle where the secondary drip begins to grow. Figure 3e zooms into a top view 3 × 5 cm area above the drip and shows an elevation contour map of the surface. This was created using data obtained from digital photogrammetry calculated using high-resolution oblique camera images (sensitive to ±0.1 mm changes; ±0.5 km scaled; see Methods section). Subsidence of the surface scaled to ~1.25 ± 0.5 km is shown in the surface elevation contour map.

In Fig. 3f, by 57.1 h, the secondary drip pulse has travelled ~1 cm deeper into the tank compared to the previous frame. The green vectors show that the descent rate has increased. The digital photogrammetry data (Fig. 3g) show that this causes an increase in surface subsidence, deepening the basin to a scaled depth of ~1.5 km ±0.5 km at its deepest point. This pulse of a secondary drip is interpreted to be causing renewed active subsidence at the surface and forming an oval shaped basin (see Supp. Fig. 3).

Figure 3h shows the model surface at the end of the experiment where there is still no significant deformation recorded across the crust associated with the drip (i.e., a horizontally unperturbed surface crustal

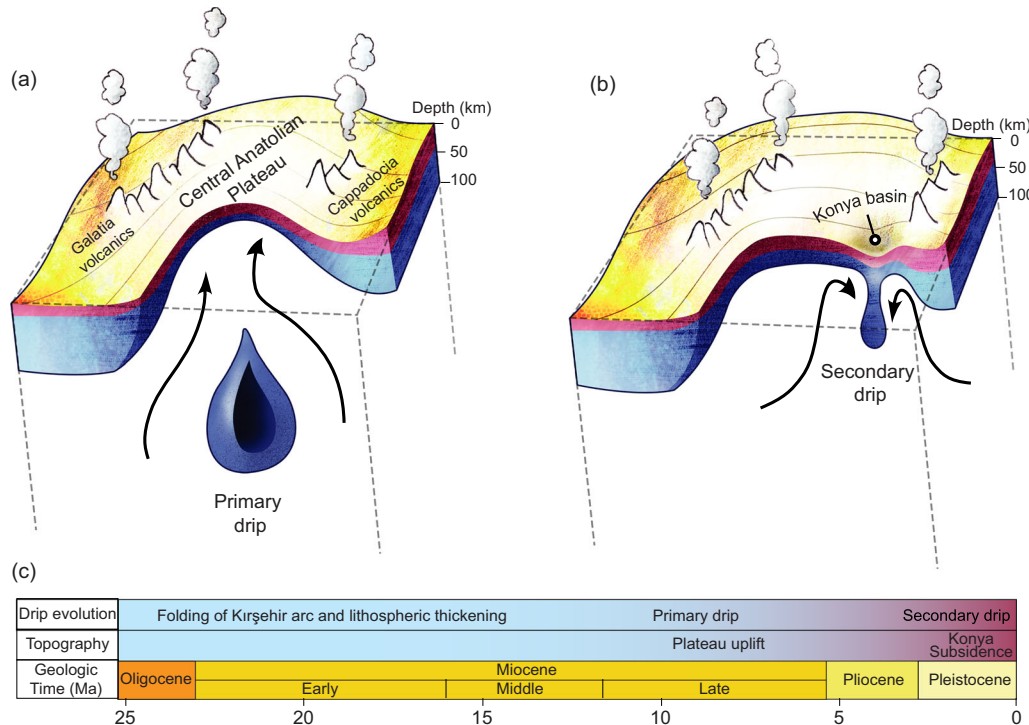

**Fig. 4 | The evolution of multiple stage lithospheric dripping process in Central Anatolia. a** Lithospheric removal via a primary drip beneath Central Anatolia causing plateau uplift since -10–8 Ma, subsequent to the shortening and thickening of the Kırşehır arc. **b** The development of a secondary drip and associated Konya basin formation. **c** Timeline of lithospheric drip and topographic evolution of Central Anatolia from 25 Ma to present.

sand layer). Figure 3i shows a contour map of the scaled vertical surface displacement, calculated by subtracting Fig. 3e from Fig. 3g. The resulting displacement field shows regions with positive vertical displacement (pink-yellow) that indicates surface uplift and regions with a negative vertical displacement (blue-black) that denotes subsidence. Notably, surface subsidence is recorded corresponding to displacement of the deepest point of the basin, outlined in the black dashed square. In summary, the active local subsidence and uplift is driven by late stage (secondary) dripping following the main lithospheric removal episode.

## Discussion

The geodynamic model presented here demonstrates how a local, oval-shaped basin can form in response to a secondary lithospheric dripping event, and this can occur without inducing crustal shortening. This model may explain the anomalous subsidence of the Konya Basin within the Central Anatolian plateau interior documented by the GNSS/INSAR data[24]. Furthermore, the geodynamic evolution of the lithospheric drip is consistent with the geological history and the geophysical constraints of the region in which present day deformation and the tectonic (seismic) activity is lowest across the Anatolian block[11,13,24,42].

The primary and secondary lithospheric dripping episodes can be understood by relating the sequence and history of tectonic events in Central Anatolia. It has been suggested that the rise of Central Anatolia over the past -10 My occurred due to dripping of the Kırşehır magmatic arc root after middle Eocene-Miocene lithospheric thickening[3,10,41] (Fig. 4a). Based on this drip tectonics model, the Galatia and Cappadocia volcanics (Figs. 1b, 4a) in Central Anatolia are formed by mantle upwelling associated with lithospheric removal, as evidenced by slow seismic wave speed anomalies[27].

The development of a secondary dripping episode below south-central Anatolia is illustrated schematically in Fig. 4b, where the seismic tomography data show a fast seismic wave speed anomaly (see

Fig. 2a). As suggested by the models, this secondary dripping event may be responsible for the current deepening of the overlying Konya Basin. In Fig. 4, the primary drip is depicted as being fully detached based on seismic tomography[27]. Whether a thin neck attaching the primary drip remains -as in the model- is uncertain owing to the resolution limitations of seismic imaging. We also note that mantle flow induced by the subduction of the Cyprus slab may have also offset the secondary drip to the south of the Kırşehır arc towards the Konya Basin.

The active subsidence of the Konya Basin within a broader framework of plateau formation is a profound illustration of how lithospheric drips can operate as an episodic process. Further, evidence suggests this process is not unique to the Konya Basin or Tethyan orogenic system in general. Based on geochemical constraints along the Cordilleran orogenic system, DeCelles et al.[40,43] and Lee et al.[44] interpret[43–45] that lithospheric foundering operates as a repetitive event through the course of magmatic arc evolution. McQuarrie et al. suggest[46] that mantle lithospheric deformation has been focused at the Eastern-Cordillera-Altiplano boundary in the Central Andes due to episodic mantle lithospheric delamination, coinciding with volcanic events. Long et al. also suggests[47] that episodes of lithospheric loss via dripping are responsible for post-rifting modification of lithospheric structure in the Central Appalachian Mountains. Dripping arc roots[48,49] and delaminating lithosphere[50,51], and their associated tectonic responses (e.g., uplift/subsidence, shortening/extension) are common processes during orogenic cycles. Our work shows how drip tectonics evolves within the Tethyan and other orogenic systems.

Basin formation within an uplifted orogenic plateau also is not unique to Central Anatolia. There are internally drained (endorheic) basins in the Altiplano-Puna and Tibetan plateaus for which their formation and evolution have been linked to lithospheric drips. For example, the Arizaro basin in the Puna region has undergone subsidence followed by uplift for the past -18 My and folding and thrust faulting[40]. In addition, the Lunpola basin of Tibet experienced

**Table 1 | Rheology, density, and thickness of analogue materials with scaling factor (SF)**

| EXP -1 | Thickness | | | Density (kg m⁻³) | | | Effective Dynamic Viscosity (Pa s) | | | φ (°) | n |
|---|---|---|---|---|---|---|---|---|---|---|---|
| | Model (mm) | Nature (km) | SF (×10⁻⁷) | Model | Nature | SF | Model (×10⁴) | Nature (×10²¹) | SF (×10⁻¹⁷) | | |
| Upper crust | 2 | 10 | 2.00 | 1100 | 3333 | 0.33 | - | - | - | 35 | - |
| Mantle Lithosphere | 20 | 100 | 2.00 | 1128 | 3418 | 0.33 | 2.92 | 1.95 | 1.50 | - | 1.01 |
| Mantle | 173 | 865 | 2.00 | 1010 | 3060 | 0.33 | 1.50 | 1.00 | 1.50 | - | 1.01 |

shortening and rapid subsidence during the Paleogene, which has been linked to a lithospheric drip[52,53]. Lithospheric drips have been invoked to explain the development of small-scale, circular basins like these, as well as the larger-scale removal of lithosphere that yielded plateau uplift in regions such as the Altiplano-Puna and Tibet[1,2,54,55]. Our findings indicate that there may be a connection between plateau uplift and basin formation events through the evolution of primary and secondary lithospheric removal.

The distinguishing characteristic of the Konya Basin among these basins is that it records minor horizontal tectonic deformation. The analogue model results presented here demonstrate that while the surface is subsiding, the drip is not inducing horizontal deformation in the crust (e.g., an asymptomatic drip) (See Supp. Fig. 2). This model behaviour can explain the relative lack of deformation in the Konya Basin while still accounting for rapid basin development. Surficial level activities such as groundwater extraction may accompany the larger scale subsidence that we address in this work[56]. Our model interpretations could account for long-term tectonic deformation, specifically ~ 60 m of basin subsidence in 3 Myr. We note that other factors, such as groundwater extraction may contribute to a large portion of the high rate of current active subsidence recorded by the InSAR/GNSS measurements[24,57].

The key conclusion of this work is that basin evolution and plateau uplift may be linked in a multistage process of lithospheric removal within a large-scale orogenic plateau system. Supported by geological, geophysical, and geodetic data, our model results explain the enigmatic active subsidence of the Konya Basin amidst the rising Central Anatolian plateau interior.

## Methods
### Materials
An analogue model is a simplified, 3D model created in the laboratory using selected materials as scaled analogues for the Earth's sub-lithospheric mantle, mantle lithosphere (upper mantle) and the upper crust. The analogue model in this study was constructed using methods from[58,59]. The sub-lithospheric mantle in the model was a viscous silicone polymer - polydimethylsiloxane (PDMS). The mantle lithosphere was created using a viscous, homogeneous mixture of 70% PDMS and 30% Plasticine™ modelling clay[59]. The modelling clay was added to increase the density and viscosity of the mantle lithosphere in relation to the underlying sub-lithospheric mantle. A brittle upper crust was created using a combination of well sorted, fine-grained (500 μm, dry, solid) silica sand and (300 μm, dry, hollow) ceramic e-spheres (Envirospheres®)[59,60]. The density, viscosity and thickness of these materials are outlined in Table 1. As previously described in ref. 59, the upper crust material was estimated to have negligible cohesion[61], but the angle of internal friction (φ) of the crust was ~35° (Table 1). This angle was determined by equating the angle of repose of the upper crust material in a conical pile to the angle of internal friction[62]. Such an approach for measuring the strength of granular materials is feasible given that the sand and e-spheres are particles of uniform size, density, and moisture content[63].

The sub-lithospheric mantle (PDMS) and mantle lithosphere (PDMS + modelling clay) materials are the same as used in ref. 59 (i.e. both viscous, non-Newtonian materials) and are defined by the power law:

$$\sigma^n = \eta \dot{\varepsilon} \quad (1)$$

where $\sigma$ is stress, $\dot{\varepsilon}$ is strain rate, $\eta$ is consistency/viscosity, and $n$ a stress exponent[58,64]. A strain-controlled rheometer (Discovery HR-3 Hybrid) was used to measure the $n$ value (Table 1) and effective dynamic viscosity ($\eta_{eff}$) of all viscous materials used in the model. The rheology values in Table 1 were recorded at a strain rate of $10^{-5} s^{-1}$, an approximate rate of deformation in the experiment[59].

### Experimental setup
The analogue experiment was conducted in the Tectonophysics lab at the University of Toronto, using the experimental setup methods described by Andersen et al.[59]. The model was scaled for length, density, dynamic viscosity, time, and gravity. A 25 × 25 × 20 cm Plexiglass box was filled with recycled PDMS (containing some clay impurities) to serve as the analogue sub-lithospheric mantle and was left for ~14 days until all air bubbles had been removed, which is essential to reduce noise in the images taken during the experiment[59]. The contrast in density between the model's mantle lithosphere and sub-lithospheric mantle, creates an instability (Rayleigh-Taylor instability) between these layers, thus drips could initiate naturally during the experiment[59,65,66]. However, to ensure the position of the primary drip, a 2.5 cm hemispherical perturbation of the same material as the mantle lithosphere was inserted into the centre of the PDMS mantle until a flat surface reformed, akin to the method described in ref. 58 and Method 2 in ref. 59. After the perturbation was inserted, a 2 cm thick viscous mantle lithosphere was then placed on top of the PDMS (and perturbation) ensuring there was minimal trapped air to reduce noise and again left until a flat surface formed[59]. Viscous drag at the box boundaries was mitigated during this step by lubricating the inside of the Plexiglass box with Petroleum Jelly. Further, the box was of sufficient volume that any drag from the edges of the box did not hinder the descent of the drip or associated flow in the centre, thus an additional lubricant such as a mixture of petroleum jelly and paraffin oil was not required[67]. To simplify the dynamics of the model, the viscous lower crust was assumed to be part of the upper mantle lithosphere[59]. The brittle upper crust was sieved in an even 2 mm thick layer on top of the mantle lithosphere. Scattered black sand particles on the surface of the model served as tracers, which are required for the image correlation techniques used in the image analysis[59]. This model had a free surface at the upper crust as the plexiglass box was open at the top.

The kinematics of the experiment were recorded by two digital cameras (CCD, 11 MPx, 16 bit) (Imager ProX by LaVision GMbH, Göttingen, Germany) from above, which took images at oblique angles to the model surface and one digital camera from the side (CCD, 11 MPx, 16 bit) (Imager Pro by LaVision GMbH, Göttingen, Germany), which captured images normal to the front plane of the box (Fig. 3 demonstrates this angle)[59]. The model was illuminated from above and from the side against a black background. This created a strong contrast, so the camera only recorded motion from the model[59]. The LaVision GMbH software recorded the evolution of the model at an interval of one image every 58 s.

For the mantle material, recycled PDMS was used which included small, dispersed particles of clay. These served as tracer particles for the imaging system where relative displacement was calculated between consecutive images, and subsequently used to calculate velocity vectors in the analogue mantle[59]. The DaVis software uses 2-dimensional Particle Imaging Velocimetry, which tracks changes in the pattern of the marker particles in consecutive images using a cross-correlation technique, from which the velocity field is calculated[68]. This was the same method employed in ref. [59], and other previous studies have also used this technique or similar imaging techniques to compute mantle flow and crustal velocity fields[59,68–73]. The contrast between the black background and the instability allowed us to track the motion of the drip and the flow in the mantle, however, this means that the results show flow velocities through the volume of the entire box projected onto the front plane of the image[59].

The StrainMaster tool in the *DaVis* software uses digital photogrammetry to compute surface height and vertical displacement, which requires the cameras to take two images at oblique angles to the model surface. The two top cameras were calibrated so the relationship between the raw oblique images and real-world space was known through a calibration model. This calibration model was used together with matching the pattern between the two cameras above the model to compute a single corrected image required for the digital photogrammetry calculations of surface elevation to a precision of ±0.0945 mm (scaling to ~±0.47 km in nature)[59].

## Scaling

To compare the analogue experiments with nature, scaling relationships were established for time, length, gravity, density, and dynamic viscosity[58,59]. The length scale ($L$) was calculated based on the mantle lithosphere thickness (denoted by subscript $m$) of $l_m = 20$ mm in the model. As mentioned previously, the lithosphere beneath Central Anatolia is thin, with the LAB located at ~60–100 km depth[25,26], therefore, we chose to scale the length of the mantle lithosphere to a natural value of 100 km. The upper crust in this model was thin and corresponds in nature to only the brittle portion of the upper crust, where geophysical studies have determined a thin, below average effective elastic thickness of 6 km in Western Anatolia[74,75]. A regional Bouguer gravity anomaly map shows high gravity anomalies in Konya (Central Anatolia)[76]. The authors of this work interpret the cause of this high anomaly to be a mass of high density oceanic (ophiolitic) crust that was obducted during the closure of the Tethys ocean and subsequently covered by a layer of sediments. Our model upper crust has a high density to approximate this condition. It has a high density (1100 kg/m³) in comparison to PDMS (1010 kg/ m³). However, analogue models are limited by the materials available, especially when selecting a granular material to mimic brittle behaviour that maintains a reasonable angle of internal friction so that it does not behave like a powder with high internal friction[59,77]. Seismic tomography and modelling studies suggest that the Central Anatolian Plateau has undergone lithospheric removal and mantle uplift that is supported by slow seismic wave speed anomalies beneath a thinned lithosphere[3,27]. This interpretation is in accord with our model where the dense mantle lithosphere (and lower crust) has undergone an initial phase of dripping and removal before the secondary drip. Petrological studies argue that basaltic melts in Central Anatolia contain high-density pyroxenite, which may be responsible for the growth of the instability and subsequent drip process in Central Anatolia[78].

The geological configuration of the study area in Central Anatolia yields the following scaling relationships. Based on an average natural mantle lithosphere thickness (denoted by subscript $n$) of $l_n = 100$ km in Central Anatolia, the length scale ratio was $L = l_m/l_n = 2.00 \times 10^{-7}$. The density of the model PDMS sub-lithospheric mantle was $\rho_m = 1010$ kg m$^{-3}$. Assuming an average density of the asthenospheric mantle in nature of $\rho_n = 3060$ kg m$^{-3}$, this yields a density scaling factor of $P = \rho_m/\rho_p = 0.33$[59]. The experiments are gravity driven and the scaling ratio for gravity is $G = g_m/g_n = 1$. The measured effective dynamic viscosity of the PDMS ($\eta_m$) was $1.9 \times 10^4$ Pa s, and scaling with an approximate viscosity of the asthenosphere ($\eta_p$) of $10^{21}$ Pa s gives a viscosity scale ratio $M = \eta_m/\eta_n = 1.50 \times 10^{-17}$[59]. A dimensional analysis of $M$, $P$, $L$, and $G$ shows the time scaling factor ($T$) can be defined as $T = M/PLG = t_m/t_n = 1.82 \times 10^{-10}$[42,59]. Using this scaling factor, 1.59 h in the model is 1 My in nature. Table 1 lists the scaling factors for the model and natural Earth values.

## Data availability

The topography map was constructed using WGM2012 Earth Gravity Model[79]; https://bgi.obs-mip.fr/grids-and-models-2/). The Analogue Modelling data used in this study are available in the Figshare data repository https://doi.org/10.6084/m9.figshare.23579823.

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

## Acknowledgements

J.A. and R.N.P. were supported by an N.S.E.R.C. Discovery Grant (RGPIN-2019-06803)-RNP. O.H.G acknowledges ANATEC (ILP/International Lithosphere Programme) and the 2232 International Fellowship for Outstanding Researchers Programme of the Scientific and Technological Research Council of Turkey (118C329). The collaborative research was enabled by a TUBITAK Fellowship for Visiting Scientists 2221 programme to RNP. EŞU thanks TUBITAK-BIDEB for support by the International Postdoctoral Research Fellowship Programme (2024-2025). This research was enabled in part by support provided by Compute Ontario (compute ontario.ca) and the Digital Research Alliance of Canada (alliancecan.ca). Nevena Niagolova (BDes, MArch) made artistic contributions/design of Fig. 4. Nevena can be contacted at www.nevena.org or @nevcheart on social media platforms.

## Author contributions

J.A., O.G. and R.P. conceived the ideas and interpretations. J.A. and T.S. conducted analogue experiments and data analysis at the University of Toronto with data analysis contributed by E.Ş.U. The research enabled in part by support provided by Compute Ontario and the Digital Research Alliance of Canada. The depiction of seismic subsurface, crustal thickness, residual topography was generated by EŞU. The manuscript was written and prepared by J.A., O.G., and R.P. with comments and input included from all authors.

## Competing interests

The authors declare no competing interests.
