## [Peer Review File · Nature Communications]

Editorial Note: Figure on page 28 of this Peer Review File has been redacted as indicated to remove third-party material where no permission to publish could be obtained.

Reviewers' comments:

Reviewer #1 (Remarks to the Author):

In this manuscript, Anderson and colleagues describe the geological setting and geophysical characteristics of the actively subsiding Konya basin in the south central part of the Anatolian plateau. Based on these characteristics they argue that the basin is a dynamic topographic feature that is forming due to the pull of a gravitationally unstable drip at the base of the mantle lithosphere. This drip is interpreted to be secondary to a, presumably larger, primary drip which formed ca. 10 Ma ago, which led to (isostatic?) uplift of the broader Anatolian plateau. The authors evaluate their hypothesis by the use of a simple 3D analogue drip experiment, which does a reasonable job of replicating the size and subsidence rate of the Konya basin.

I found the idea that a small, actively subsiding, closed basin could be forming due to the action of a secondary gravitational instability within a still uplifting orogenic plateau to be particularly noteworthy and original. The idea is tested by a novel experimental setup capable of measuring both flow in the asthenospheric mantle and surface topography in response to the dripping instability. For these reasons I think the work is worthy of publication after fairly major revisions, which are recommended to make the communication of results and their interpretation more robust. I have provided an annotated PDF, which the authors should use for their revisions and only summarise the major points in my review below.

1. The manuscript is generally well written and the figures are mostly of good quality. However, there are some sections of text that have rather awkward phrasing or odd or unsuitable word choices, which are indicated in the PDF. Likewise, the figure captions could be clearer and include more detailed explanation of the figure contents, which are quite complicated in places.

2. I struggled a bit with what is known about the current state of the Anatolian lithosphere and its tectonic evolution. It is not clear until one reads the Methods section what the current thickness of the lithosphere is (i.e., it is quite thin with a thick crust and a thinned mantle lithosphere). This and how so much lithosphere came to be removed needs to be more clearly explained early on in the manuscript, particularly in the context of the broad-scale plateau uplift that started ca 10 Ma ago. It needs to be spelled out more clearly that this initial lithosphere removal event is attributed to a (presumably much larger) primary drip (of the Korsehir magmatic arc root?) and the the current high velocity feature in the lithosphere below the Konya basin and the associated surface subsidence are relatively late features attributed to a smaller, secondary drip.

3. In a similar manner, the early history/behaviour in the drip experiment is not presented so the context of what happens between 50.6 and 57.1 hours is rather unclear. It would be very beneficial to the reader to know what happened in the first 50.6 hours of the experiment. When did the primary drip form? How large was it? How fast did it descend? How large was the associated surface basin? Did uplift occur after the primary drip stopped being active? How did the timing of these events and their scales match up with Anatolian tectonics? There seems to be a missed opportunity here.

4. Following on from this, I found the rationale for the setup of the experiment a bit confusing. The

lithospheric thickness appears to have been chosen based on modern day constraints from geophysics, which is after removal of a lot of lithosphere by the primary drip, which drove regional plateau uplift around 10 Ma and thereafter. Surely, a more appropriate starting condition should have the original crust and mantle lithosphere in place so the experiment can evolve to form a primary drip, have lithosphere removed, and then develop a secondary drip? The thickness of the lithosphere in the experiment at 57.1 hours could then be compared to present-day values in Anatolia as another check.

5. I had a few questions and suggested improvements in the Methods section. For example, the density of the upper crust reported in Table 1 seems implausibly high and if correct will have significantly unrealistic body force effects. The explanations of the PIV and digital photogrammetry techniques could be clearer and improved. There is important geological/geophysical background information hiding in the Methods section that needs to be front and centre in the main body of the text.

6. Lastly, why did the secondary drip form in the first place? Do secondary drips form in all of these types of experiments or was the one presented here special in some way? Some information/discussion along these lines would help to broaden the impact/relevance of the contribution.

Reviewer #2 (Remarks to the Author):

This paper starts with an interesting idea, that a basin formed on the Central Anatolian Plateau owes its existence to downward motion in the mantle underneath the crust. The idea is appealing, but the question perhaps is whether the authors have done enough to make the idea convincing. The paper begins with a description of the relevant observations, in particular the evidence from recently published InSAR measurements which apparently show the surface of the Konya Basin in the Central Anatolian Plateau subsiding, from a seismic tomography model in which a relatively high-velocity volume in the uppermost mantle beneath the basin is evident, and from a model of residual topography that depends on gravity and crustal thickness estimates. The authors support their idea with a description of a fluid dynamical experiment in which they observe downward flows in a tank of material called PDMS.

The tank observations relate to the period following an initial Rayleigh-Taylor instability in which an upper dense layer forms a central blob which sinks through the mantle. However the authors do not relate the basin formation to the initial instability of the layer, but rather to a “late-stage” event in which a secondary drip forms and downward motion is accelerated. What exactly causes this late-stage event is not explained. It appears to be a relatively minor perturbation in the diameter of the vertical column of dense material, perhaps consequent on the prior slowing of the flow when the initial blob hit the floor of the tank. I am wondering why, if this small perturbation in the flow can have such a significant effect on the surface, what would have been the effect of the initial instability? And why not attempt to explain the basin formation in terms of a more recent primary blob formation event?

What the authors have to say about velocity is a bit hard to pin down. The experimental observations seem to relate primarily to the velocity of the drip in the mantle at a scaled depth of about twice the thickness of the mantle. They actually say that the flow velocities are integrated throughout the volume of the box, but surely they don't mean the entire volume of the box. So where exactly is the velocity

measured ? At some depth, or depth range ? There are also estimates of surface subsidence velocity from the InSAR and these can be related to surface measurements from the experiments for which they infer deflections of about +/- 1.5 km, and claim a precision of 95 microns (0.5 km scaled). In Fig 3 we see plots of surface deflection (made complicated and essentially unusable by using colour and 3D projection at the same time) and in Fig 3e of the subsidence rate distribution at a time that is late in the experiment. I find it difficult to know what to take away from this experiment when we are given only snippets of information about it, but lack for example a clear understanding of how the experiment is proceeding in time. We are also shown in Fig 3 grey rectangles that purport to be raw PIV camera images. Why ? They show nothing useful to the reader, but obscure the surface deflection plots, which should be shown without distortion on a 2D plot.

The authors also make a simplifying assumption of using a relatively dense crustal layer in the experiment. This can have a significant dynamical effect, especially when it comes to computing surface deflection and deflection rate. This is a major difference from the actual geophysical problem where crustal buoyancy is more significant, sufficient to cast in doubt any kind of quantitative scaling of the surface deflection inferred from the tank experiments.

In summary, I like the geological concept and the tank experiments are interesting but could be explained in more detail. However, the use of a relatively high density, thin crust in the experiments, seems like a problem in that it minimizes crustal buoyancy compared to the Earth. Consequently I don't find that the authors are making a convincing case for how to quantitatively apply the results from the tank experiments to the geological problem. I feel that these problems need to be addressed in more detail before the paper would be acceptable.

Reviewer #3 (Remarks to the Author):

Dear Editor,

I have read the manuscript entitled Multistage lithospheric drips control active basin formation within an uplifting orogenic plateau submitted by Andersen et al. The paper strives to elucidate the formation of the Konya Basin within the Central Anatolian Plateau. They suggest that the ongoing 5 cm/yr subsidence rate based on InSAR and GNSS analysis is a consequence of deep-rooted geophysical phenomena such as the descent of a lithospheric drip based on fast seismic wave anomaly beneath the basin. This manuscript is not ready to publish in your journal and needs to be more elaborated with local geology and geomorphology. I am a geomorphologist; therefore, I will particularly emphasize a few critical geomorphic points that authors must consider.

1. The time span that InSAR and GNSS analysis covers is very short from a geological point of view; however, lithospheric dripping takes millions of years. If you combine these two data, there is a considerable time gap between them and it would be better if authors look at local geology and geomorphology to fill this gap. For instance, did they check the inclination of bedding of sedimentary rocks? Do they tilt to the Konya Basin or not? If we consider 5 cm/yr subsidence, there must be significant tilting, although it is an asymptomatic drip.

2. Similarly, local geomorphology. In our paper (Melnick et al., 2017), we studied paleolake shorelines in Tuz Gölü and Konya Basin. These are the shorelines belonging to LGM (20 ka) Lake. As you know, the

shorelines are the best geomorphic markers to quantify vertical deformations. It would be really useful for the authors if they checked these data and correlated them with their model. If we consider 5 cm/yr subsidence, all these paleo shorelines need to be strongly deformed, which is not the case.

3. Authors suggest there is no crustal deformation within the Konya Basin. Still, in our paper (Melnick et al., 2017), we showed the 12 m vertical displacement along the Sultanhanı Fault and a surface rupture that might be produced by an earthquake magnitude 6.5 or higher. Additionally, an earthquake ML 5.1 occurred on August 1 2023, in Konya within the subsidence circle of the paper, indicating obvious crustal deformation. It is rare but exists.

4. I think this comment is the most critical because the authors did not consider the Konya Basin's karstic characteristics and the human effect. If you look at the subsidence pattern, the centre of the Konya Basin is stable and not subsiding. This area corresponds to Bozdağ Mountain National Park, with no agriculture. All the rapidly subsiding areas overlap with karstic plains that are intensively cultivated. Since the area is semiarid, the farmers pump more underground water every year, and this extraction of groundwater gives rise to subsidence of the land surface, like in San Joaquin Valley, California, in the 1950-60s. This is a well-known process in the region, and every year we have news about the formation of sinkholes within the Konya basin because of the drop in underground water level.

I recommend that the Authors consider these issues.

For these reasons, I found the manuscript not ready to publish in Nature Communications.

Response to Reviewer Comments

Line numbers correspond to the original document that was submitted for review, not the newly revised manuscript.

Response to Reviewer 1

In this manuscript, Andersen and colleagues describe the geological setting and geophysical characteristics of the actively subsiding Konya basin in the south central part of the Anatolian plateau. Based on these characteristics they argue that the basin is a dynamic topographic feature that is forming due to the pull of a gravitationally unstable drip at the base of the mantle lithosphere. This drip is interpreted to be secondary to a, presumably larger, primary drip which formed ca. 10 Ma ago, which led to (isostatic?) uplift of the broader Anatolian plateau. The authors evaluate their hypothesis by the use of a simple 3D analogue drip experiment, which does a reasonable job of replicating the size and subsidence rate of the Konya basin.

- 1) I found the idea that a small, actively subsiding, closed basin could be forming due to the action of a secondary gravitational instability within a still uplifting orogenic plateau to be particularly noteworthy and original. The idea is tested by a novel experimental setup capable of measuring both flow in the asthenospheric mantle and surface topography in response to the dripping instability. For these reasons I think the work is worthy of publication after fairly major revisions, which are recommended to make the communication of results and their interpretation more robust. I have provided an annotated PDF, which the authors should use for their revisions and only summarise the major points in my review below.
- 2) I struggled a bit with what is known about the current state of the Anatolian lithosphere and its tectonic evolution. It is not clear until one reads the Methods section what the current thickness of the lithosphere is (i.e., it is quite thin with a thick crust and a thinned mantle lithosphere). This and how so much lithosphere came to be removed needs to be more clearly explained early on in the manuscript, particularly in the context of the broad-scale plateau uplift that started ca 10 Ma ago. It needs to be spelled out more clearly that this initial lithosphere removal event is attributed to a (presumably much larger) primary drip (of the Kırşehir magmatic arc root?) and the current high velocity feature in the lithosphere below the Konya basin and the associated surface subsidence are relatively late features attributed to a smaller, secondary drip.

Response: In the revised manuscript we elaborate on the tectonic evolution of the two-stage lithospheric dripping process at central Anatolia where the initial event produces plateau uplift, and the later process causes active subsidence of the Konya basin. Specifically, we have added in the Introduction and Geophysical Constraints sections passages of several sentences (with appropriate references) that explain how the folded Kırşehir arc formed an arc root and mantle lithosphere removal causing plateau uplift. We further clarify that the drip that produced plateau uplift is a primary drip. In addition, we completely redrafted Figure 4—the schematic diagram that illustrates how the two-stage dripping process works and revised the associated text in the Discussion to explain more clearly.

To address the reviewer's other comment, we added a profile in figure 2 that shows the current crustal thickness. We also added a sentence to the 'Geophysical constraints on the evolution of the Konya basin' section that explains that the lithospheric thickness beneath the plateau is in the range of 60 km – 100 km (Vinnik et al., 2014; Kind et al, 2015).

- 3) 3. In a similar manner, the early history/behaviour in the drip experiment is not presented so the context of what happens between 50.6 and 57.1 hours is rather unclear. It would be very beneficial to the reader to know what happened in the first 50.6 hours of the experiment. When did the primary drip form? How large was it? How fast did it descend? How large was the associated surface basin? Did uplift occur after the primary drip stopped being active? How did the timing of these events and their scales match up with Anatolian tectonics? There seems to be a missed opportunity here.

Response: In the revised manuscript we added new photos in Figure 3 and explanations that describe the early behavior in the drip experiments—i.e., in the first 50.6 hours (with images at 10h and 25h after initiation of the experiment). This illustrates the early-stage development of lithospheric drip where large -scale removal caused plateau uplift. We emphasize that the purpose of this work is not to focus on the plateau formation in the early stage; rather it aims to explain anomalous active subsidence in the Konya basin (secondary drip). Due to the nature of geodynamic experimental work, a given experiment cannot explain the geological evolution of a region in terms of complete events while matching their scales.

We thank the reviewer for pointing out this opportunity and we have enhanced explanations of the primary and secondary removal events to clarify the full scope of tectonic events in central Anatolia.

- 4) Following on from this, I found the rationale for the setup of the experiment a bit confusing. The lithospheric thickness appears to have been chosen based on modern day constraints from geophysics, which is after removal of a lot of lithosphere by the primary drip, which drove regional plateau uplift around 10 Ma and thereafter. Surely, a more appropriate starting condition should have the original crust and mantle lithosphere in place so the experiment can evolve to form a primary drip, have lithosphere removed, and then develop a secondary drip? The thickness of the lithosphere in the experiment at 57.1 hours could then be compared to present-day values in Anatolia as another check.

Response: In the revised manuscript we clarified that lithospheric thickness estimate is not selected based on current geophysical constraints where it is in the range of 60-100 km (Vinnik et al 2014; Kind et al., 2015), therefore we chose 110 km for the lithospheric thickness as a reasonable approximation for orogenic areas/magmatic arcs. To advance further our two-stage drip hypothesis we elaborated on the final figure (Figure 4) where a range of geological, geophysical, and petrological constraints are interpreted.

- 5) I had a few questions and suggested improvements in the Methods section. For example, the density of the upper crust reported in Table 1 seems implausibly high and if correct will have significantly unrealistic body force effects. The explanations of the PIV and

digital photogrammetry techniques could be clearer and improved. There is important geological/geophysical background information hiding in the Methods section that needs to be front and centre in the main body of the text.

Response: In the revised manuscript we explain why our initial crustal density is at the higher end compared to regular densities for the crust. This is because the density of the crust underneath central Anatolia, specifically the Konya basin, is high as inferred from geophysical analyses (i.e., +0.25 kg/m³ Bouguer gravity anomaly) (Aydın & İşseven, 2018). This estimate is supported by the area being located above a Tethyan suture, represented by (higher density) ophiolitic rocks. We have explained this condition in the revised manuscript (Methods section) to clarify our setup. Description of the primary model parameters has also been added to the main text, as recommended by the reviewer. We have improved the explanations of PIV and digital photogrammetry as well.

- 6) Lastly, why did the secondary drip form in the first place? Do secondary drips form in all of these types of experiments or was the one presented here special in some way? Some information/discussion along these lines would help to broaden the impact/relevance of the contribution.

Response: In the Discussion section of the revised manuscript, we expand on this point to describe that secondary drips can be a natural consequence of ongoing lithospheric removal/dripping processes. Our experience from both numerical and analogue experiments demonstrates that once the lithosphere becomes unstable and the removal is instigated, multistage dripping may occur. Depending on rheological and density variations, the process may be more pronounced and here we relate this in the context of orogenic cyclicality. In the revised manuscript, we cited other cases where such repetitive dripping has been interpreted for various regions around the globe, based on geological and geophysical data.

Reviewer 1 kindly provided an annotated pdf document with many phrasing, grammar, and punctuation corrections. We have implemented those suggested changes to improve the clarity of the work. We have provided responses to the comments on this document in this response letter.

- 1) Line 50-53: Suggest you break this up into 2 sentences - otherwise it is hard to follow. Also, what do you mean by "outstanding" - outstanding in what way?

Response: This sentence was broken up into 2 sentences and the word outstanding was removed from the new manuscript.

- 2) Line 55: Do you need to say "based on stratigraphic analysis"?

Response: This statement was removed from the manuscript.

- 3) Line 75: should be "oxygen stable isotope ratios within."

Response: This wording has been changed in the new manuscript

- 4) Line 88: What does this mean? Structural observations? Normally one would do surface mapping and look for features like slickenlines, kinematic indicators and offset markers to make such deductions. "Subsurface data" implies either data collected underground in mines, or by drilling or from geophysics (3D seismic/potential field data, etc).

Response: The phrase “subsurface data” now reads as “structural, stratigraphic, and sedimentological data” to specify the type of data we are referencing.

- 5) Line 89: This sounds like the fault is not well understood or that it has a complex multiphase history.

Response: The fault does have a complex multiphase history. In the revised manuscript, we have modified this explanation to clarify the main kinematic behaviour of the fault (with reference to Cemen et al., 1999; Fernandez-Blanco, 2013).

- 6) Line 93: Sorry, but this hasn't been defined or described yet, so how are we supposed to know what and where it is? There is something labelled Tuz Gölü in Fig. 1b but there is no explanation about what it is.

Response: Fig 1b now specifies that there is a Tuz Gölü basin and a Tuz Gölü fault, which are labelled. We have also modified the explanation in this passage, so that specific reference to the Tuz Golu basin is now gone.

- 7) Line 95: between? How can something begin from the late Cretaceous TO the Miocene?

Response: We have changed the wording of this in the updated manuscript and it now reads as “between the Late Cretaceous and the Miocene epochs.”

- 8) Line 96: preserves a record of long-term?

Response: This has been changed in the updated manuscript to read as “The basin preserves a record of long-term sedimentation”

- 9) Line 98: This doesn't follow from the first part of the sentence

Response: This sentence was broken up into two sentences for clarity.

- 10) Line 116: Use ca. for time - tilda for things like dimensions, etc

Response: We have corrected this in the revised manuscript.

- 11) Line 117: Do you need to repeat this information here?

Response: We have kept this information to emphasize this puzzling phenomenon in the Konya Basin. It's an important part of the observations that we will be returning to, and want to make this clear to readers.

- 12) Line 118: This sentence doesn't really add much” and “Why recycling? What is being recycled?

Response: In the revised manuscript, we modified the term recycling and explain how multistage removal may be an important mechanism for orogenic cycle.

- 13) Line 125: This and the following text seem to imply that the base of the lithosphere is at ~50 - 60 km depth, which seems very shallow for an orogenic plateau with a ~40 km thick crust.

Response: We clarified this lithospheric structure based on geophysical studies with new references (Vinnik et al. 2014; Kind et al. 2015). These authors estimate that the lithosphere beneath Central Anatolia is thin, and the base of the lithosphere is ~60 km – 100 km. We have elaborated on this more precise information in the geophysical constraints section of the revised manuscript.

- 14) Line 126: Can you please explain what you mean by "undulation" and "protrude" here? Also what is the significance of the larger high Vs anomaly north of the Konya basin under the Kirsehir arc?

Response: We have modified this entire discussion to clarify the structural geometry of the lithosphere in the region, with reference to Figure 2. To explain the high Vs anomaly to the north of the Konya basin, we added a sentence regarding the previous geodynamic history of the Central Anatolian plateau where folding of the arc formed a thickened (higher density) arc root beneath the Kirsehir arc.

15) Line 127: North of? Above? "In front" seems like an odd word choice

Response: The wording in this sentence has been changed to read as “north of the interpreted Cyprus Slab”.

16) Line 129: What do you mean by anomaly in this context?” “According to Fig. 2, the small circular region of thicker crust seems to be offset from the centre of the basin.

Response: In this region within and around Konya, we see a thickened crust ~40-44 km, thicker than the surrounding crust in the plateau at ~34 km. There is thick crust at ~40 km in the basin with thicker anomalies at the peripheries. We have revised the manuscript to state this.

17) Line 136: ???

Response: We have removed the statement “(i.e., down going mantle flow support on topography)” from the revised manuscript. The description of residual topography has been re-written and clarified.

18) Line 143 -146: See previous comment on the location/depth of the LAB. I think it is critical here to define where the base of the lithosphere is.

Response: As described above, the lithosphere beneath Central Anatolia is 60 – 100 km thick based on seismic data from Vinnik et al. (2014) and Kind et al. (2015) (as implicitly recommended by the reviewer, we added explanation/reference of this in the revised manuscript.) The base of the anomaly referred to in this passage is 60 km, consistent with these available data.

19) Line 164-166: But does the modelling also account for simultaneous plateau uplift that you also claim to be driven by a lithospheric drip?

Response: Yes, it does, although this primary drip event is not the main focus of this paper. We have made substantial revisions to the manuscript to explain this. In the revised Figure 3 we now include a depiction of the full drip process evolution. In Figure 3h, we show vertical displacement. The main point we emphasize is that the basin created by the secondary drip is deepening (illustrated by the dashed box). However, surrounding this area, the vertical displacement is positive (i.e., the surface is uplifting as a plateau). Over the duration of the initial stages of the experiment, the surface progressively uplifted (consistent with the plateau uplift from Gogus et al., 2017).

20) Line 169: Some context is required for the reader to understand what is meant by late stage versus early-intermediate stages. It would therefore be useful to describe what happens in this kind of drip experiment over time, particularly since such information is not presented in the methods section either.

Response: We agree with the reviewer that this was not well explained in the original manuscript. We have updated Figure 3 to show the first 25h of the experiment where the primary drip has almost touched the bottom of the tank (Figure 3a-c). It emphasizes the larger scale of the drip in comparison to the secondary drip.

- 21) Line 177: There is no context/explanation provided for what is meant by "raw" camera image here, in the methods section or the figure caption, so it just comes across as jargon. Why not just say what it is, which is an oblique digital image of the model surface?
Response: As recommended, we revised the description of the modeling method to be clearer about how camera images are used to estimate the surface topography associated with the model.
- 22) Line 181: Why not just say "deformation" here? Also, importantly, was there any deformation observed of the upper crust in the early and intermediate stages of the experiment?
Response: For clarity, we have changed "strain localization" to deformation. Figure 3c shows an image of the surface of the experiment at 25h where there was no crustal shortening or significant deformation earlier in the experiment. We explain this in the revised manuscript.
- 23) Line 193: It is important to also outline what happened at early and intermediate stages. For example, how do you know what components of subsidence can be attributed to the initial drip versus the secondary drip?
Response: In general, the primary drip caused an overall surface uplift, whereas the secondary drip is causing a late-stage subsidence. We have clarified this in the revised manuscript with the modifications to Figure 3 and modified written passages.
- 24) Line 196: Should be i.e., = that is. Not e.g., = for example
Response: Thank you, we have changed this to i.e.,
- 25) Line 197: How much of this is noise and how much is real? No errors have been quantified/presented.
Response: This figure has been revised. Rather than vertical velocity, we use vertical displacement in the updated Figure 3h. The error on the camera measurements is ± 0.0945 mm; this is given in the Methods section of the revised manuscript.
- 26) Line 203: This was never documented, described or explained
- 27) Line 214: These were never adequately explained.
- 28) Response (Points 25 and 26): The substantial revisions made to Figure 3 and the text now describe the primary drip progression. These explanations are distributed through the Introduction and Modelling sections of the revised manuscript. In addition, we refer to earlier work (Gogus et al., 2017) that focuses specifically on this type of primary drip event.
- 29) Line 215-217: This requires some further explanation/clarification. Active dripping doesn't cause surface uplift, so what one can guess here is that the surface uplift is an isostatic response to removal of the magmatic arc root by a dripping mechanism.
Response: This sentence has been changed in the updated manuscript to read as "It has been suggested that the rise of Central Anatolia over the past ~10 My occurred due to dripping of the Kırşehir magmatic arc root after middle Eocene-Miocene lithospheric thickening (Göğüş et al., 2017; Meijers et al., 2018; McPhee et al., 2022) (Fig. 4a)". This will clarify how this uplift is occurring.
- 30) Line 220: This would have been good to know a lot earlier on.

In the revised manuscript, we now explain in the Introduction (i.e., much earlier) about these events: seismic evidence for lithospheric thinning (Fichtner et al., 2013) and interpretations for lithospheric removal and plateau uplift.

31) Line 226: towards?

Response: we have changed this in the updated manuscript to read “towards the Konya Basin”.

Comments on Table 1

32) These thickness values seem to be very different to the Anatolian plateau and Konya basin

Response: The brittle upper crust is modeling the purely brittle part of the upper crust which has a below average thickness of 6 km in Western Anatolia, so we chose 10 km for simplicity. We assumed the viscous lower crust was part of the mantle lithosphere. In total the lithospheric thickness beneath Anatolia is 60 km – 100 km so we chose the higher end and have a mantle lithospheric thickness of 100 km.

33) This is a very strange density to use for the continental crust. It is even higher than the asthenosphere!

Response: This is the density of a fine grained, silica sand that we used as the model upper crust. We tested two different crust materials (sand and silica spheres/e-spheres) in a series of experiments, and neither crust seemed to influence the outcome of the experiment but rather the outcome was influenced by how the drip was initiated and the rheology of the mantle lithosphere. Aydin & İşseven (2018) report high Bouguer gravity values in the Konya region of Turkey. They attribute this to a portion of dense oceanic crust (ophiolitic) that was obducted during the closure of the Tethys Ocean and subsequently then covered by sediments burying it to a depth of ~4.8 km. Therefore, we believe that a dense crust is reasonable for this model.

34) This table reports more than just "rheology"

Response: The title of the table has been changed in the updated manuscript to “Table 1 | Rheology, density and thickness of analogue materials with scaling factor (SF).”

35) Line 284: Surely it was also added to change the rheology/viscosity?

Response: Thank you for mentioning this. The clay did also increase the viscosity. This has been updated in the manuscript to read “to increase the density and viscosity of the mantle lithosphere.”

36) Line 286: Please check that the density reported in Table 1 is correct. It doesn't seem likely for this mix of materials.

Response: Thank you for highlighting this error. The density in the table was that of a fine-grained silica sand. The experiment shown here had an upper crust of sand rather than silica spheres and e-spheres. We have updated the manuscript so that the correct material is listed.

37) Line 287: Not just rheological...

Response: We have removed “rheological properties”, and the updated manuscript now reads as “The density, viscosity and thickness of these materials are outlined in Table 1.”

38) Line 308-310: Since you use only 2 layers, this is a pretty extreme approximation!

Response: We agree that this is an approximation. This approach is considered to be fairly standard in scaled laboratory experiments.

39) Line 356: You've now said this 3 times.

Response: We have changed this paragraph in the updated manuscript so as not to repeat information.

40) Line 357: You could say this more simply. PIV tracks changes in patterns of particles using a cross-correlation technique, from which the velocity field is calculated.

Response: We have updated how we explain this in the new manuscript.

41) Line 360: Specifically the Strainmaster routine within DaVis

Response: The updated manuscript now reads as “The Strainmaster routine in the DaVis software uses digital photogrammetry to compute surface height and vertical displacement”.

42) Line 362: This could be said more simply. Also, "raw" image has no context because you don't explain that DaVis computes a corrected image from the two oblique raw images. You also need explain the the cameras take images at oblique angles of the model surface, which is required for the photogrammetry calculations.

Response: We have updated this section to describe that the overhead cameras are oblique to the model surface and that we use these images with a calibration model to produce a single corrected image that is required for the photogrammetry calculations.

43) Line 369: Please explain all subscripts

Response: We have clarified the manuscript to state the subscript m is for model values and subscript n is for natural Earth values

44) Line 369-371: It would have helped to learn this much earlier in the paper

We have added description of this and the Fichtner et al. (2013) reference to the Introduction.

45) Line 371: But this is presumably the thickness after the initial removal of mantle lithosphere. Surely the model needs to start with the original lithosphere thickness?

Response: Thanks for pointing this again. In the revised manuscript we clarified this initial condition in various places.

46) Line 380-382: There is a problem here because you model should really start with the original crustal and lithospheric thickness before the formation of the primary drip and removal of mantle and crust alluded to here.

Response: Thanks for pointing this again. In the revised manuscript we clarified this initial condition in various places.

47) Line 386: I assume you mean asthenospheric mantle. Upper mantle could also mean lithospheric mantle.

Response: Yes, we have revised it to read as “asthenospheric mantle”.

Comments on Figure 1

48) Line 676: Not clear how these are shown on this map

Response: We have removed “Major plate boundaries” from the caption. Figure 1a shows Tectonic motions of Anatolia.

49) Line 679: Sorry, but isn't this a shaded relief topographic map?

Response: The caption now reads as “Shaded relief topographic map of Central Anatolia”.

50) Line 681: The faults are actually white lines. Only the half arrows indicating sense of shear are yellow.

Response: We have changed “yellow” to “white” in the updated manuscript and added that yellow arrows indicate the sense of shear.

51) Line 683: Labelling is confusing because only the inset is labelled (c) - The whole figure is (c) and the inset zooms in to the Konya Basin

Response: Figure 1c) is meant to be the zoomed inset. We show the entire map to show where that data came from.

52) Line 684: Define V_{up} (mm/yr) please.

Response: We have updated the caption to read as “vertical velocities (V_{up}) in mm/yr”.

Comments on Figure 2

53) Line 686: What is the dashed white line? LAB??

Response: We have changed this figure in the updated manuscript. The new caption explains that the dashed white line is the LAB.

Response to Reviewer 2

1) This paper starts with an interesting idea, that a basin formed on the Central Anatolian Plateau owes its existence to downward motion in the mantle underneath the crust. The idea is appealing, but the question perhaps is whether the authors have done enough to make the idea convincing. The paper begins with a description of the relevant observations, in particular the evidence from recently published InSAR measurements which apparently show the surface of the Konya Basin in the Central Anatolian Plateau subsiding, from a seismic tomography model in which a relatively high-velocity volume in the uppermost mantle beneath the basin is evident, and from a model of residual topography that depends on gravity and crustal thickness estimates. The authors support their idea with a description of a fluid dynamical experiment in which they observe downward flows in a tank of material called PDMS.

Response: Thank you for the positive comments on our manuscript. We have made numerous changes in the manuscript to clarify and strengthen our explanation of this new idea. Figure 4 has been revised to a more sophisticated cartoon graphic that clearly illustrates our idea that the Konya Basin is subsiding due to a secondary drip. We have also provided references to instances in other locations where researchers also suspect this is occurring in the central Andes, Tibet and Appalachian mountains. Figure 3 was updated to show how the primary drip progression and how a secondary drip developed in the model following the initial dripping incident.

2) The tank observations relate to the period following an initial Rayleigh-Taylor instability in which an upper dense layer forms a central blob which sinks through the mantle. However, the authors do not relate the basin formation to the initial instability of the layer, but rather to a “late-stage” event in which a secondary drip forms and downward motion is accelerated. What exactly causes this late-stage event is not explained. It

appears to be a relatively minor perturbation in the diameter of the vertical column of dense material, perhaps consequent on the prior slowing of the flow when the initial blob hit the floor of the tank. I am wondering why, if this small perturbation in the flow can have such a significant effect on the surface, what would have been the effect of the initial instability? And why not attempt to explain the basin formation in terms of a more recent primary blob formation event?

Response: In the revised manuscript, we expanded on this point with a new Figure 4 where the primary drip produced uplift and formation of the central Anatolian plateau approximately 10-8 Ma and the subsequent late-stage blob/drip produced active subsidence of the Konya basin. Most of that connection between the two-stage dripping process was interpreted in the context of the geodynamic evolution of central Anatolia. The model evolution shows that the primary drip hits the bottom of the tank much earlier (25 hours after the initiation of the experiment), therefore, the formation of the late-stage event is not caused by a boundary effect. Please see new Figure 3 for illustration of the full dripping process.

We hope that with the new figures 3 and 4 we have a more thorough illustration and description of the process: the first drip with large scale material removal/drip is linked to plateau uplift; and the subsequent one with smaller scale dripping is related to active basin formation.

- 3) What the authors have to say about velocity is a bit hard to pin down. The experimental observations seem to relate primarily to the velocity of the drip in the mantle at a scaled depth of about twice the thickness of the mantle. They actually say that the flow velocities are integrated throughout the volume of the box, but surely they don't mean the entire volume of the box. So where exactly is the velocity measured? At some depth, or depth range? There are also estimates of surface subsidence velocity from the InSAR and these can be related to surface measurements from the experiments for which they infer deflections of about +/- 1.5 km, and claim a precision of 95 microns (0.5 km scaled). In Fig 3 we see plots of surface deflection (made complicated and essentially unusable by using colour and 3D projection at the same time) and in Fig 3e of the subsidence rate distribution at a time that is late in the experiment. I find it difficult to know what to take away from this experiment when we are given only snippets of information about it, but lack for example a clear understanding of how the experiment is proceeding in time. We are also shown in Fig 3 grey rectangles that purport to be raw PIV camera images. Why? They show nothing useful to the reader, but obscure the surface deflection plots, which should be shown without distortion on a 2D plot.

Response: Based on the reviewer's comments, we made appreciable modifications to Figure 3:

- 1) We show the entire evolution of the drip experiment, including the earlier stages, so that the full experiment (and cycle of the drip process) may be better understood.
- 2) The surface velocities are now omitted because they can not be directly reconciled with INSAR based constraints owing to various uncertainties. We rather show the amount

of vertical displacements between consecutive time slices to quantify subsidence of a Konya-type basin and surrounding uplift.

3) We also deleted the grey rectangles showing the raw PIV camera images. In the revised figure 3, we demonstrate more thorough analyses of topographic evolution of the lithospheric drip process.

- 4) The authors also make a simplifying assumption of using a relatively dense crustal layer in the experiment. This can have a significant dynamical effect, especially when it comes to computing surface deflection and deflection rate. This is a major difference from the actual geophysical problem where crustal buoyancy is more significant, sufficient to cast in doubt any kind of quantitative scaling of the surface deflection inferred from the tank experiments.

Response: In the revised manuscript, we explain why our initial crustal density is at the higher end compared to known densities for the crust. This is because the density of the crust underneath central Anatolia, specifically the Konya basin, seems to be elevated as inferred from geophysical analyses (i.e., +0.25 kg/m³ Bouguer gravity anomaly) (Aydın and İşseven, 2018). This is supported by the area being located above a Tethyan suture, represented by (higher density) ophiolitic rocks. We added this explanation and reference to this work in the revised manuscript to clarify the issue. In this model, we used fine-grained, silica sand as the model upper crust. We tested two different crust materials (sand and silica spheres/e-spheres) in a series of experiments, and neither crust seemed to influence the outcome of the experiment but rather the outcome was influenced by how the drip was initiated and the rheology of the mantle lithosphere.

- 5) In summary, I like the geological concept and the tank experiments are interesting but could be explained in more detail. However, the use of a relatively high density, thin crust in the experiments, seems like a problem in that it minimizes crustal buoyancy compared to the Earth. Consequently, I don't find that the authors are making a convincing case for how to quantitatively apply the results from the tank experiments to the geological problem. I feel that these problems need to be addressed in more detail before the paper would be acceptable.

Response: We thank the reviewer for their positive comments on our work. In the revised manuscript, we explain (including a new figure 3 and 4) how our results may account for the active subsidence of the Konya basin in central Anatolia. The comparison of the modeling results with geophysical observables in Figure 2 reconciles the geodynamics with the geological problem but uncertainties both in the model and the constraints can make the direct comparison challenging. However, this work can address an enigmatic concept in the tectonics of Anatolia and is probably applicable to other areas around the globe.

Response to Reviewer 3

- 1) Dear Editor,

I have read the manuscript entitled Multistage lithospheric drips control active basin formation within an uplifting orogenic plateau submitted by Andersen et al. The paper

strives to elucidate the formation of the Konya Basin within the Central Anatolian Plateau. They suggest that the ongoing 5 cm/yr subsidence rate based on InSAR and GNSS analysis is a consequence of deep-rooted geophysical phenomena such as the descent of a lithospheric drip based on fast seismic wave anomaly beneath the basin. This manuscript is not ready to publish in your journal and needs to be more elaborated with local geology and geomorphology. I am a geomorphologist; therefore, I will particularly emphasize a few critical geomorphic points that authors must consider.

Response: We thank the review for their comments. In the revised manuscript, we improved the explanation of the two-stage lithospheric drip concept based on geological information through new figure 4. The new figure 3, which shows subdued tectonic deformation as well as vertical displacement amounts and patterns of uplift and subsidence can address important points in geomorphology and tectonics.

- 2) The time span that InSAR and GNSS analysis covers is very short from a geological point of view; however, lithospheric dripping takes millions of years. If you combine these two data, there is a considerable time gap between them and it would be better if authors look at local geology and geomorphology to fill this gap. For instance, did they check the inclination of bedding of sedimentary rocks? Do they tilt to the Konya Basin or not? If we consider 5 cm/yr subsidence, there must be significant tilting, although it is an asymptomatic drip.

Response: We agree with the reviewer that lithospheric dripping is a rather long-term tectonic behavior (millions of years) and we recognize that InSAR/GNSS is a measure of present day subsidence. Nevertheless, to support the lithospheric drip hypothesis with scaled laboratory experiments we also incorporated other constraints including crustal thickness variations, seismic tomography, non-isostatic (residual topography) topography inferred from gravity inversions. These geological and geophysical observations provide information on the tectonic scales the reviewer is raising (the “time gap”), but this doesn’t preclude us from using InSAR/GNSS as a constraint as well. We note that tilting of the bedding of sedimentary layers may not necessarily reflect lithospheric scale geological processes (e.g., dripping, subduction); it can also be formed by local tectonic activities, such as normal faulting.

- 3) Similarly, local geomorphology. In our paper (Melnick et al., 2017), we studied paleolake shorelines in Tuz Gölü and Konya Basin. These are the shorelines belonging to LGM (20 ka) Lake. As you know, the shorelines are the best geomorphic markers to quantify vertical deformations. It would be really useful for the authors if they checked these data and correlated them with their model. If we consider 5 cm/yr subsidence, all these paleo shorelines need to be strongly deformed, which is not the case.

Response: In the revised manuscript we added a discussion to clarify the point raised by the reviewer here (e.g., comparative scales between the experiments and nature). We note that the Melnick et al. (2017) work was cited in the introduction of the original submission, but we have also added related discussion in the revised Introduction. It is not our finding of the high rate of active surface subsidence (5 cm/yr); rather it has been estimated by INSAR and GNSS studies. While our experimental findings are in accord with this subsidence controlled by lithospheric instabilities/drip crust-uppermost mantle

scale, and are consistent with geophysical and geological studies, more localized geomorphic features cannot develop in these experiments. Owing to the limitations of the experimental work (essentially the challenge of dealing with extreme widths in spatial and temporal scales) we cannot simulate exact development of the local geological processes, such as erosion, deposition, groundwater extraction in these tectonic-scale models.

- 4) Authors suggest there is no crustal deformation within the Konya Basin. Still, in our paper (Melnick et al., 2017), we showed the 12 m vertical displacement along the Sultanhanı Fault and a surface rupture that might be produced by an earthquake magnitude 6.5 or higher. Additionally, an earthquake ML 5.1 occurred on August 1 2023, in Konya within the subsidence circle of the paper, indicating obvious crustal deformation. It is rare but exists.

Response: As the active fault map of Anatolia as well as earthquake database below show, central Anatolia is associated with subdued tectonic activity unlike the east and the west of the country. We agree that the region is not completely absent of tectonic deformation; rather—and as the reviewer notes--it is rare, and this region represents a relatively quiet tectonic deformational region of Anatolia.

[Figure Redacted]

- 5) I think this comment is the most critical because the authors did not consider the Konya Basin's karstic characteristics and the human effect. If you look at the subsidence pattern, the centre of the Konya Basin is stable and not subsiding. This area corresponds to Bozdağ Mountain National Park, with no agriculture. All the rapidly subsiding areas overlap with karstic plains that are intensively cultivated. Since the area is semiarid, the farmers pump more underground water every year, and this extraction of groundwater gives rise to subsidence of the land surface, like in San Joaquin Valley, California, in the 1950-60s. This is a well-known process in the region, and every year we have news about the formation of sinkholes within the Konya basin because of the drop in underground water level.

Response: In the revised manuscript, we added this point into the Discussion section—that such surficial level activities (hydrological basin) may accompany larger-scale subsidence in the Konya region. This is an important point, and indeed the groundwater extraction may account for the very high InSAR/GNSS rates that our models can't fully explain, but it does not invalidate our findings. Our proposal for lithospheric drip hypothesis is in good agreement for a range of observations (e.g thicker crust above subsiding region, high speed seismic anomaly) where deep tectonic effects drive the anomalous topography.

REVIEWER COMMENTS

Reviewer #1 (Remarks to the Author):

This is a revised version of an original manuscript submitted by Anderson and colleagues. I have read the new manuscript and the responses to the three reviewers comments. I note also that the comments by Reviewer 2 raised many of the same concerns as in my original review. Overall, I am satisfied that the authors have addressed most of the substantive comments provided in all three reviews and I congratulate the authors on a much improved manuscript.

That said, I think that a number of minor improvements could be made before publication. Firstly, I have provided an annotated PDF with a few editorial suggestions and comments. Secondly, I have several trailing questions about the work summarised below:

1. As requested, the authors have provided some additional information about the first 25 hours of the drip experiment, when the "primary" drip was most active. However, it would also be useful to provide additional information on the surface topography evolution before the secondary drip started to form. This is because it is impossible to tell from Fig. 3e what aspects of the surface topography are related to the secondary drip and what is "inherited" from the behaviour of the primary drip. This could be handled by adding some more information and figures to the Supplement.
2. I like Figure 4 as it captures the main concepts nicely. However, it doesn't quite match with the experiment presented, in which the "secondary" drip is effectively a pulse of new material that has flowed into the tail of the primary drip. I.e., both drips are co-located, while Figure 4 portrays the secondary drip as as a new R-T instability that forms in a different location, after the primary drip has detached from the mantle lithosphere. This is not what happened in the experiment reported, although such truly secondary drips have been observed in other experiments (e.g., Pysklywec & Crude, 2004). Some comment/explanation on this in the text needs to be added to build confidence in the applicability of the experimental results to the Konya Basin.
3. I still have some questions about material properties in the experiments. Firstly, what makes the mantle lithosphere denser than the asthenosphere to drive a Rayleigh-Taylor instability in the first place? Is there an explanation for what made this happen in the Anatolian mantle? And can the scaled lithospheric mantle density of 3418 kg/m^3 in the experiment be justified? Secondly, the density of the dry quartz sand (1100 kg/m^3) used for the upper crust in the experiment seems unusually low. This is because silica (SiO_2) has a density of 2.65 kg/m^3 (this cannot be changed) such that the bulk density of pure quartz sand ($= f(\text{silica grain density} + \text{air density})$) is typically $\sim 1400 \text{ kg/m}^3$ for most reasonable porosities ($= \text{vol \% sand vs vol\% air}$). So the silica sand reported in Table 1 must either have an enormously high porosity or it has been reduced by adding some lower density particles.

Reviewer #2 (Remarks to the Author):

First a complaint about the management of this manuscript – whether a failing of the authors or of

Nature Communications - Figure 1 is in the wrong orientation and half of the map is truncated and not visible. If you are sending a manuscript for review, why can't it include the figures in a coherent package with a consistent page size, even if not in-line. You could make the reviewer's task a little simpler. Also, why aren't the figures for the Supplementary Information also included with the text in the reviewer package?

I reviewed an earlier version of this manuscript. I think the authors have taken on board one of my key criticisms by describing in more detail the experiment on which the paper is based. The physical process now makes more sense to a reader, even if the conditions that establish the analog experiment are somewhat different to those that presumably established the actual geological environment. The context of the geological environment differs most notably from the experiment because of the Cyprus slab, which is imaged as the major feature in the seismic tomography (Fig 2a), but has no parallel in the analog experiment. The authors ignore this difference, but still the experiment has something to say about the process of secondary drips forming, and the relevance to the Konya basin seems plausible. The study therefore should justify a publication – if they rectify the following problem:

The authors have also retreated to some degree from the claims that their surface deflection measurements can be related to observed surface subsidence in the Konya basin. However they have not retreated far enough in my opinion. Put simply, it is misleading and irrelevant that a rate of 5 cm/yr subsidence observed today in the basin is mentioned in the 2nd sentence of the abstract, and given a paragraph of the text, as if it has something to do with the mechanism that they propose here. Reviewer 3 point 5 offers a more likely explanation of the geodetic observation. The scaled rate of subsidence indicated in their own experiment (roughly 60 m in 3 Myr) is orders of magnitude slower than the GNSS rate. They should be looking for geological / sedimentological evidence of subsidence rate for the process they invoke (they mention sedimentation, but don't infer a rate from it), but they cite the geodetic evidence that is relevant to a different process, and then paper over the inconsistency with a statement like: "Groundwater extraction may account for part of the high InSAR/GNSS displacements, but it does not invalidate our findings". This sort of misleading and inconsistent logic does not belong in the paper.

Other minor comments:

line 376, 377: I guess they mean integrated along the line of sight rather than over the volume of the whole box, but even so, is it really integrated, or are the particles illuminated on a particular plane orthogonal to the line of sight ?

The figures are improved, but still in a couple of cases the figures are not particularly suitable for providing easily accessible quantitative information. Why for example are Fig 2c and 2d shown with different colour scales, when they nominally show the same quantity computed by two different groups. We could compare them more easily if you use the same colour scale. It is also not clear how to interpret what is shown in Fig 3c and 3h. What are we supposed to infer from those figures ? The caption does not help.

Reviewer #4 (Remarks to the Author):

The manuscript has already been through 1 round of review, and I'm asked to assess the authors' responses to the referees' previous concerns, as well as commenting on the robustness, novelty and importance of this manuscript for Nature Communication.

I read the revised submission in accordance with the comments by 3 reviewers who provided extensive and stimulating contributions as being experts on the relevant fields of geophysics, geomorphology, geology and experimental modelling. The authors have considered all comments and implemented meticulously to improve their submission as the geodynamic model presented in the paper demonstrates how an internally drained sedimentary basins within the interior of orogenic plateaus originates and such a basin can form in response to a secondary lithospheric dripping event without crustal shortening. To support this phenomenon, the authors attempt to use an analogue laboratory model to experiment how an active formation in a particular basin (i.e. Konya basin of central Anatolia) may be accounted for by the descent of a lithosphere drip causing local subsidence. Eventually the authors interpret the development of the basin through a secondary drip pulse concurrent with broad plateau uplift caused by a larger-scale lithospheric drip. Therefore, the research reveals that basin evolution and plateau uplift may be linked in a multistage process of lithospheric removal during episodic development of orogenic systems. Eventually the authors attempted to modify the term recycling and explain how multistage removal may be an important mechanism for orogenic cycle.

Overall the model presented in this study is consistent with the actual Konya basin case as well as in accordance with the other examples elsewhere World and enhance our understanding on the interaction between surface and subsurface dynamics which is still enigmatic, then need to be tested as in present case. Finally, as noted by the authors, underscores what might be an overlooked multistage process of lithospheric removal within a large-scale orogenic system.

In the meantime, I particularly focussed on the comments by the Reviewer 3, who emphasised the effect of karstic processes due to groundwater extraction may account for high InSAR/GNSS rates forming the subsidence. The authors noted this point and included satisfactory justification on this possibility; it is meaningful, needs to be considered but doesn't invalidate their proposal at all.

Apart from all comments by 3 reviewers, I have a concern on the Fig.3 (i) on which there are several other depressions even deeper than the secondary drip-induced indicated by dashed rectangle. In the overall experimental model, there is a single primary drip followed by a secondary trip that causes subsidence which is attributed to as an analogy for the Konya basin. So, what would be the explanation for the other depressions (even deeper) around? The authors may note these multi-depressions caused by secondary drip, and may favour a depression complex rather than single depression attributed to a single drip event. In fact, basins are not single depressions, but constituted by several depressions eventually forming a basin with several depressions in it.

As a consequence, I found the idea of two stage drip hypothesis robust and novel to better understand surface-subsurface interactions and explain basin formation along that. This is indeed a new approach and the current case is another example to support multi-stage drop mechanism causing basin

formation. Therefore, it is important for Nature Communication as it is present in revised and improved form.

REVIEWER COMMENTS

Reviewer #1 (Remarks to the Author):

This is a revised version of an original manuscript submitted by Anderson and colleagues. I have read the new manuscript and the responses to the three reviewers comments. I note also that the comments by Reviewer 2 raised many of the same concerns as in my original review. Overall, I am satisfied that the authors have addressed most of the substantive comments provided in all three reviews and I congratulate the authors on a much improved manuscript.

Response: We thank the reviewer for their positive statements.

That said, I think that a number of minor improvements could be made before publication. Firstly, I have provided an annotated PDF with a few editorial suggestions and comments. Secondly, I have several trailing questions about the work summarised below:

1. As requested, the authors have provided some additional information about the first 25 hours of the drip experiment, when the "primary" drip was most active. However, it would also be useful to provide additional information on the surface topography evolution before the secondary drip started to form. This is because it is impossible to tell from Fig. 3e what aspects of the surface topography are related to the secondary drip and what is "inherited" from the behaviour of the primary drip. This could be handled by adding some more information and figures to the Supplement.

Response: As recommended by the reviewer, in the supplementary section of the revised manuscript, we added relevant figures of evolution of surface topography for the primary drip development. We created a new plot that shows topography at 10h/5.0 My and 25h/12.6 My (corresponding to Figure 3a&b), and for comparison 50.6h/25.4 My and 57.1h/28.7 My (corresponding to Figure 3e&g but plotted at a different scale). We added relevant text in the Supplementary section to explain this new figure (LINES 451-460). Characteristic to lithospheric drip models, in these earlier stages there is a surface subsidence above the downwelling, and around it, there is uplift. We note in the manuscript that the experiment presented here does not account for the complete geodynamic evolution of central Anatolia, since 10 Ma. The foundations of our interpretations are also based on a previous contribution (Göğüş et al., 2017) where the origin of plateau uplift was discussed (LINES 76-78)

2. I like Figure 4 as it captures the main concepts nicely. However, it doesn't quite match with the experiment presented, in which the "secondary" drip is effectively a pulse of new material that has flowed into the tail of the primary drip. I.e., both drips are co-located, while Figure 4 portrays the secondary drip as as a new R-T instability that forms in a different location, after the primary drip has detached from the mantle lithosphere. This is not what happened in the experiment reported, although such truly secondary drips have been observed in other experiments (e.g., Pysklywec & Cruden, 2004). Some comment/explanation on this in the text needs to be added to build confidence in the applicability of the experimental results to the Konya Basin.

Response: We appreciate the reviewer's comment on this. The new illustration in Figure 4 was intended to show how primary and secondary drips are related and is a simplified view on drip tectonics in central Anatolia. Namely, consecutive drips as in the figure were interpreted based on the seismic tomography model of Fichtner et al. (2013, EPSL) where the first one is supposedly detached (therefore causing plateau uplift) and the secondary one we believe to be driving the active subsidence of the Konya basin. In the revised manuscript, based on the suggestions of the reviewer, we clarified this point in the discussion section.

“In Figure 4 the primary drip is depicted as being fully detached on the basis of seismic tomography (Fichtner et al 2013). Whether a thin neck attaching the primary drip remains-as in the model-is uncertain owing to the resolution limitations of seismic imaging. We also note that mantle flow induced by the subduction of the Cyprus slab may have also offset the secondary drip to the south of the Kırşehir arc towards the Konya Basin.” (LINES 241-245).

We also cited Pysklywec and Cruden (2004, G-cubed), which recognizes the development of secondary mantle lithospheric instabilities in analogue experiments.

“Notably, a secondary drip event (Fig. 3d-h) develops following the primary drip (similar to the phenomenon observed in Pysklywec and Cruden, 2004).” (LINES 189-190).

In the original manuscript we explained why there may be an offset between the primary and secondary drips. Namely, in the response above we mention that this may be due to the flow induced by the Cyprus slab which did not exist in our experiments. We also note that the new Supplementary Figure 3 suggested by the Reviewer shows that there is a shift in the center of the depressions induced by the primary and secondary drips, and make a note of that in the explanation of the figure.

3. I still have some questions about material properties in the experiments. Firstly, what makes the mantle lithosphere denser than the asthenosphere to drive a Rayleigh-Taylor instability in the first place? Is there an explanation for what made this happen in the Anatolian mantle? And can the scaled lithospheric mantle density of 3418 kg/m^3 in the experiment be justified?

Response: Petrological studies argue that basaltic melts in central Anatolia contain high density pyroxenite which may be the reason for the growth of the instability and hence a drip process in central Anatolia (Kürkçüoğlu and Yürür, 2022, Geochemistry). The presence of garnet pyroxenites has been used to account for lithospheric foundering in other areas, including the Sierra Nevada (e.g., Saleeby et al., 2003, Tectonics; Lee and Anderson, 2015, Sci. Bull.). In the revised manuscript, we have added explanation and reference to these potential petrological considerations on the development of the instability in central Anatolia (LINES 396-398).

Secondly, the density of the dry quartz sand (1100 kg/m^3) used for the upper crust in the experiment seems unusually low. This is because silica (SiO_2) has a density of 2.65 kg/m^3 (this cannot be changed) such that the bulk density of pure quartz sand (silica grain density + air density) is typically $\sim 1400 \text{ kg/m}^3$ for most reasonable

porosities (=vol % sand vs vol% air). So the silica sand reported in Table 1 must either have an enormously high porosity or it has been reduced by adding some lower density particles.

Response: We thank the reviewer for raising this point as the initial manuscript did not clearly explain the properties of crustal materials in the model. A review of our laboratory notes reveal that the material used for the upper crust was not pure silica sand. The upper crust was a combination of silica sand and e-spheres (hollow ceramic spheres), which was the same material used in analogue models of Santimano & Pysklywec (2020). This explains the lower density listed in Table 1. The manuscript has been revised accordingly to more clearly explain the nature of the model crust (LINES 301-303).

Reviewer 1 kindly provided a pdf document with further comments. Below are responses to these comments. The line numbers correspond to the line numbers in the pdf document.

1. Line 28: "Crustal thickening" is not a noun. This should read: reveal thickened crust OR reveal a region of crustal thickening.....

Response: The revised manuscript now reads as "thickened crust".

2. Line 31: Should be: calculated residual topography...

Response: The revised manuscript now reads as "calculated residual topography".

3. Line 188: Should really say: gravitationally unstable lithosphere.

Response: The revised manuscript now reads as "gravitationally unstable lithosphere".

4. Line 209: How do you define the basin width here? A certain topographic contour value? I am asking this question because the 0 km contour does not seem to have moved much between frames.

Response: We are defining the basin width on the 0 km contour. The Reviewer is right that any widening (movement of the 0 km contour) is too subtle to notice. In the revised manuscript we have removed reference to "widening" of the basin here.

5. Line 210: This statement is hard to assess because no information on the state of the surface topography before the secondary drip feature develops is provided.

Response: In the revised manuscript, we have included a new supplementary figure (Supplementary Figure 3) which shows topography at earlier stages of the experiment in relation to the primary drip. We have added a reference to this new figure here and have added a sentence to reference the figure in the manuscript (LINES 186-188).

6. Line 304: The density in Table 1 (1100 kg/m³) seems too low for silica sand (usually ~1400 kg/m³). This suggests that something else has been added to reduce the density or that the density reported in the table is incorrect.

Response: This comment was addressed above. We consulted our lab notes and realized that the Reviewer was correct. There were e-spheres added to the sand, which reduced the density of the crust.

7. Line 340: This sentence seems out of place here.

Response: This sentence has been removed from the revised manuscript.

8. Line 369: Imaging.

Response: Particle Image Velocimetry now reads as “Particle Imaging Velocimetry” in the revised manuscript.

9. Line 406: Does this need to be repeated here?

Response: This sentence was removed from the revised manuscript.

Reviewer #2 (Remarks to the Author):

First a complaint about the management of this manuscript – whether a failing of the authors or of Nature Communications - Figure 1 is in the wrong orientation and half of the map is truncated and not visible. If you are sending a manuscript for review, why can't it include the figures in a coherent package with a consistent page size, even if not in-line. You could make the reviewer's task a little simpler. Also, why aren't the figures for the Supplementary Information also included with the text in the reviewer package?

Response: It is unfortunate that the Reviewer received the half of the figure because we properly uploaded each figure into the system according to the instructions. It seems like a problem occurred somewhere with the document handling to cut off part of the figure. Sorry for the inconvenience.

I reviewed an earlier version of this manuscript. I think the authors have taken on board one of my key criticisms by describing in more detail the experiment on which the paper is based. The physical process now makes more sense to a reader, even if the conditions that establish the analog experiment are somewhat different to those that presumably established the actual geological environment. The context of the geological environment differs most notably from the experiment because of the Cyprus slab, which is imaged as the major feature in the seismic tomography (Fig 2a), but has no parallel in the analog experiment. The authors ignore this difference, but still the experiment has something to say about the process of secondary drips forming, and the relevance to the Konya basin seems plausible. The study therefore should justify a publication – if they rectify the following problem:

The authors have also retreated to some degree from the claims that their surface deflection measurements can be related to observed surface subsidence in the Konya basin. However they have not retreated far enough in my opinion. Put simply, it is misleading and irrelevant that a rate of 5 cm/yr subsidence observed today in the basin is mentioned in the 2nd sentence of the abstract, and given a paragraph of the text, as if it has something to do with the mechanism that they propose here. Reviewer 3 point 5 offers a more likely explanation of the geodetic observation. The scaled rate of subsidence indicated in their own experiment (roughly 60 m in 3 Myr) is orders of magnitude slower than the GNSS rate. They should be looking for geological / sedimentological evidence of subsidence rate for the process they invoke (they mention sedimentation, but don't infer a rate from it), but they cite the geodetic evidence that is relevant to a different process, and then paper over the inconsistency

with a statement like: “Groundwater extraction may account for part of the high InSAR/GNSS displacements, but it does not invalidate our findings”. This sort of misleading and inconsistent logic does not belong in the paper.

Response: We appreciate the Reviewer’s comments on this. Based on their comments, in the revised manuscript we modified our explanations—specifically backing away from a direct comparison with the present day (GNSS derived) subsidence rate. To avoid being misleading, we removed the rate of subsidence from the abstract. Unfortunately, to the best of our knowledge, there does not exist published sedimentological data that constrains the rate of subsidence in the Konya basin. We removed some of the previous text that made direct comparisons and added a new discussion with references for the hydrological effects of subsidence in the Konya basin. Based on the Reviewer’s comments and suggestions here, the relevant passage now reads as follows:

“Our model interpretations account for long-term tectonic deformations, specifically ~ 60 m of basin subsidence in 3 Myr. We note that other factors, such as groundwater extraction may contribute to a large percentage of the high rate of current active subsidence recorded by the InSAR/GNSS measurements (Caló et al., 2017; Weiss et al., 2020) (LINES 280-283).

Other minor comments:

Line 376, 377: I guess they mean integrated along the line of sight rather than over the volume of the whole box, but even so, is it really integrated, or are the particles illuminated on a particular plane orthogonal to the line of sight?

Response: Based on the Reviewer’s comments, in the revised manuscript, we clarified our statement as such:

“The contrast between the black background and the instability allowed us to track the motion of the drip and the flow in the mantle, however, this means that the results show flow velocities through the volume of the entire box projected onto the front plane of the image.” (LINES 362-365)

The figures are improved, but still in a couple of cases the figures are not particularly suitable for providing easily accessible quantitative information. Why for example are Fig 2c and 2d shown with different colour scales, when they nominally show the same quantity computed by two different groups. We could compare them more easily if you use the same colour scale. It is also not clear how to interpret what is shown in Fig 3c and 3h. What are we supposed to infer from those figures ? The caption does not help.

Response: As suggested by the Reviewer, Figure 2c and 2d have been adjusted so that they now use the same colour scale. For Figure 3c and 3h, we rewrote the caption and added manuscript text to clearly explain what these figure frames demonstrate.

Namely, they show that there is no visible horizontal deformation recorded at the surface sand/crust layer. We also link this with the relevant Supplementary Figure 2 that explains symptomatic vs. asymptomatic drips. As explained in the text, such asymptomatic behaviour seems to agree with the evolution of the Konya basin in central Anatolia.

Reviewer #4 (Remarks to the Author):

The manuscript has already been through 1 round of review, and I'm asked to assess the authors' responses to the referees' previous concerns, as well as commenting on the robustness, novelty and importance of this manuscript for Nature Communication.

I read the revised submission in accordance with the comments by 3 reviewers who provided extensive and stimulating contributions as being experts on the relevant fields of geophysics, geomorphology, geology and experimental modelling. The authors have considered all comments and implemented meticulously to improve their submission as the geodynamic model presented in the paper demonstrates how an internally drained sedimentary basins within the interior of orogenic plateaus originates and such a basin can form in response to a secondary lithospheric dripping event without crustal shortening. To support this phenomenon, the authors attempt to use an analogue laboratory model to experiment how an active formation in a particular basin (i.e. Konya basin of central Anatolia) may be accounted for by the descent of a lithosphere drip causing local subsidence. Eventually the authors interpret the development of the basin through a secondary drip pulse concurrent with broad plateau uplift caused by a larger-scale lithospheric drip. Therefore, the research reveals that basin evolution and plateau uplift may be linked in a multistage process of lithospheric removal during episodic development of orogenic systems. Eventually the authors attempted to modify the term recycling and explain how multistage removal may be an important mechanism for orogenic cycle.

Overall the model presented in this study is consistent with the actual Konya basin case as well as in accordance with the other examples elsewhere World and enhance our understanding on the interaction between surface and subsurface dynamics which is still enigmatic, then need to be tested as in present case. Finally, as noted by the authors, underscores what might be an overlooked multistage process of lithospheric removal within a large-scale orogenic system.

In the meantime, I particularly focussed on the comments by the Reviewer 3, who emphasised the effect of karstic processes due to groundwater extraction may account for high InSAR/GNSS rates forming the subsidence. The authors noted this point and included satisfactory justification on this possibility; it is meaningful, needs to be considered but doesn't invalidate their proposal at all.

Apart from all comments by 3 reviewers, I have a concern on the Fig.3 (i) on which there are several other depressions even deeper than the secondary drip-induced indicated by dashed rectangle. In the overall experimental model, there is a single primary drip followed by a secondary trip that causes subsidence which is attributed to as an analogy for the Konya basin. So, what would be the explanation for the other depressions (even deeper) around? The authors may note these multi-depressions caused by secondary drip, and may favour a depression complex rather than single depression attributed to a single drip event. In fact, basins are not single depressions, but constituted by several depressions eventually forming a basin with several depressions in it.

Response: We appreciate the Reviewer's positive comments on the work. In Figure 3i (change in elevation between 50.6 h and 57.1 h) the localized depressions on the margins of the main basin shown in the centre are only transient expressions of surface elevation differences at these specific times (i.e., showing up by calculating a difference in elevation with these data to create 3i). Since they are just transient motions in the model, they do not play a role in the overall basin development—i.e., the evolution of the main subsidence in the central part of the model.

We appreciate the Reviewer's comment that basins can develop as depression complexes, but in this case these particular features in the model are not sufficiently long-lasting to modify the local basin development.

As a consequence, I found the idea of two stage drip hypothesis robust and novel to better understand surface-subsurface interactions and explain basin formation along that. This is indeed a new approach and the current case is another example to support multi-stage drop mechanism causing basin formation. Therefore, it is important for Nature Communication as it is present in revised and improved form.

REVIEWERS' COMMENTS

Reviewer #1 (Remarks to the Author):

This is the third version of this manuscript that I have been asked to review. The authors have addressed all of my previous concerns and suggestions and I consider the manuscript ready for publication. Congratulations on a well-presented and thought-provoking paper.